# FBXO7 ubiquitinates PRMT1 to suppress serine synthesis and tumor growth in hepatocellular carcinoma

Li Luo[1,2,3,5], Xingyun Wu[1,5], Jiawu Fan[1,5], Lixia Dong[1], Mao Wang[1], Yan Zeng[1], Sijia Li[1], Wenyong Yang[4], Jingwen Jiang ®[1] ✉ & Kui Wang ®[1] ✉

Cancer cells are often addicted to serine synthesis to support growth. How serine synthesis is regulated in cancer is not well understood. We recently demonstrated protein arginine methyltransferase 1 (PRMT1) is upregulated in hepatocellular carcinoma (HCC) to methylate and activate phosphoglycerate dehydrogenase (PHGDH), thereby promoting serine synthesis. However, the mechanisms underlying PRMT1 upregulation and regulation of PRMT1-PHGDH axis remain unclear. Here, we show the E3 ubiquitin ligase F-box-only protein 7 (FBXO7) inhibits serine synthesis in HCC by binding PRMT1, inducing lysine 37 ubiquitination, and promoting proteosomal degradation of PRMT1. FBXO7-mediated PRMT1 downregulation cripples PHGDH arginine methylation and activation, resulting in impaired serine synthesis, accumulation of reactive oxygen species (ROS), and inhibition of HCC cell growth. Notably, FBXO7 is significantly downregulated in human HCC tissues, and inversely associated with PRMT1 protein and PHGDH methylation level. Overall, our study provides mechanistic insights into the regulation of cancer serine synthesis by FBXO7-PRMT1-PHGDH axis, and will facilitate the development of serine-targeting strategies for cancer therapy.

To support survival and sustained proliferation, cancer cells commonly reprogram metabolic patterns to fulfill their high anabolic demands and maintain redox balance during cancer progression[1,2]. Among them, the de novo serine synthesis, a side-branch of glycolysis, is often hyperactivated in many cancer cells[3]. The serine biosynthesis starts with the conversion of the glycolytic intermediate 3-phosphoglycerate (3-PG) to 3-phosphohydroxypyruvate (3-PHP) catalyzed by the first rate-limiting enzyme phosphoglycerate dehydrogenase (PHGDH), and continues with two enzymatic reactions to generate serine[4,5]. Serine can then be catalyzed by serine

hydroxymethyltransferase 1/2 (SHMT1/2) to form glycine, which is a major one-carbon donor for one-carbon metabolism[6]. Through coupling one-carbon metabolism, serine synthesis generates biomass, energy, and intracellular reductants required for cancer cell growth and proliferation[7]. In addition to serine biosynthesis, cancer cells can alternatively uptake serine and glycine from the environment to replenish the intracellular serine and glycine pool[8]. Recent preclinical studies have reported that targeting serine metabolism by blocking endogenous serine synthesis and/or limiting the exogenous import of serine and glycine reveals potent anticancer effects[9,10], which

[1]West China School of Basic Medical Sciences & Forensic Medicine, State Key Laboratory of Biotherapy, West China Hospital, West China School of Public Health and West China Fourth Hospital, Sichuan University, 610041 Chengdu, P. R. China. [2]Center for Reproductive Medicine, Department of Gynecology and Obstetrics, West China Second University Hospital, Sichuan University, 610041 Chengdu, P. R. China. [3]Key Laboratory of Birth Defects and Related Diseases of Women and Children (Sichuan University), Ministry of Education, 610041 Chengdu, P. R. China. [4]Department of Neurosurgery, Medical Research Center, The Third People's Hospital of Chengdu, The Affiliated Hospital of Southwest Jiaotong University, The Second Chengdu Hospital Affiliated to Chongqing Medical University, 610014 Chengdu, P.R. China. [5]These authors contributed equally: Li Luo, Xingyun Wu, Jiawu Fan. ✉e-mail: jjwcn@foxmail.com; kuiwang@scu.edu.cn

inspires us to pursue the underlying mechanism of serine synthesis in cancer.

Protein arginine methylation mediated by protein arginine methyltransferases (PRMTs) has been emerging as a common and vital post-translational protein modification in cancer pathogenesis[11]. Among the reported nine PRMT family members, PRMT1 is a type I protein arginine methyltransferase that catalyzes the mono-methylation and asymmetrical di-methylation of various substrates. The PRMT1-mediated methylation of these substrates has been reported to regulate diverse cellular processes, including transcription, mRNA splicing, DNA damage and repair, metabolism remodeling, and signal transduction[12]. It has been reported that the protein level of PRMT1 is upregulated in hepatocellular carcinoma (HCC) to promote tumor progression, and its high level correlates with poor clinical outcomes in HCC patients[13,14]. Recently, we found that the aberrant high protein level of PRMT1 methylates phosphoglycerate dehydrogenase (PHGDH) and enhances its catalytic activity, thereby potentiating serine synthesis, alleviating oxidative stress, and promoting HCC growth[15]. However, how PRMT1 is upregulated in cancer remains poorly understood.

F-box protein 7 (FBXO7) is an SCF (SKP1/cullin-1/F-box protein)-type E3 ubiquitin ligase belonging to cullin-RING ligase (CRLs) family[16]. Different from other F-box proteins, FBXO7 exhibits both SCF-dependent and independent functions. FBXO7 plays a role in mitophagy[17], proteasomal regulation[18], cell cycle progression[19], and immune evasion[20] in an SCF-independent manner. In terms of its SCF-dependent function, FBXO7 interacts with and ubiquitinates specific substrates, thus regulating several cellular processes, including mitophagy[21], proteasomal assembly[22], and glycolysis[23]. Recent studies have demonstrated that FBXO7 is deregulated in several cancers, such as lung cancer, colon cancer and endometrial cancer[19,24]. However, whether FBXO7 plays a role in HCC remains unclear.

In this study, we show that FBXO7 binds and ubiquitinates PRMT1, leading to the ubiquitin-mediated degradation of PRMT1. Downregulation of FBXO7 in HCC elevates PRMT1 protein level, thereby inducing PHGDH methylation and activation, potentiating serine synthesis, ameliorating oxidative stress, and promoting HCC cell growth. Our results indicate the FBXO7–PRMT1–PHGDH axis as a critical mechanism for the regulation of serine metabolism in HCC.

## Results

### FBXO7 interacts with PRMT1

We and others have revealed that the protein level of PRMT1 is frequently upregulated in HCC to promote tumor growth[13–15,25]. To investigate the mechanism of PRMT1 upregulation in HCC, qPCR analysis was performed to examine the mRNA level of *PRMT1* in 20 HCC tissues and adjacent normal liver tissues. Interestingly, no obvious change was observed in *PRMT1* mRNA level in HCC tissues compared with normal liver tissues (Supplementary Fig. 1a). We thus speculated that other mechanisms, such as suppressed ubiquitin-proteasome degradation, might be involved in PRMT1 upregulation. To test this hypothesis, we treated Huh7 and PLC/PRF/5 cells with two different proteasome inhibitors (MG132 and bortezomib), and observed a dose-dependent increase of PRMT1 protein level following MG132 or bortezomib treatment (Supplementary Fig. 1b, c), suggesting that the protein level of PRMT1 can be regulated by ubiquitin-meditated degradation.

To identify the key E3 ubiquitin ligase for PRMT1 degradation, a screening workflow based on different databases was designed (Fig. 1a). We identified 904 PRMT1-interacting proteins from the Bio-GRID database. The 904 PRMT1-interacting candidates were then overlapped with the GOCC ubiquitin ligase complex gene set from the GSEA-MSIGDB database, and 31 proteins were identified as components of the ubiquitin ligase complex. Among them, 5 proteins were further identified from the MGI database as regulators of K48-linked

ubiquitination. Because PRMT1 promotes HCC growth, the E3 ligase, which downregulates PRMT1, theoretically plays a tumor-suppressive role in HCC. We, therefore, screened good prognosis genes for HCC patients from the TCGA database and found that only FBXO7 expression predicted favorable survival among the 5 candidates (Fig. 1b). In addition, we performed label-free quantitative proteomics analysis to identify the interacting partners of FLAG-PRMT1 and found that FBXO7 was the only E3 ligase among the top 10 PRMT1-interacting candidates (Supplementary Fig. 2a).

To determine whether FBXO7 functions as an E3 ligase for PRMT1 degradation, we examined their physical interaction. As shown in Supplementary Fig. 2b, c, the interaction between FBXO7 and PRMT1 was verified by co-IP assay in HEK293T cells with ectopic expression of FLAG-FBXO7 and GFP-PRMT1. Reciprocal co-IP analysis confirmed the binding of endogenous FBXO7 with PRMT1 in human Huh7 and PLC/PRF/5 HCC cells (Fig. 1c, d). This interaction was largely abrogated when *PRMT1* was knocked down (KD) using two different shRNA constructs (Fig. 1e, f) or two different siRNA sequences (Supplementary Fig. 2d, e). Moreover, FBXO7 was colocalized with PRMT1 in both Huh7 and PLC/PRF/5 cells, further confirming the interaction of FBXO7 with PRMT1 (Supplementary Fig. 2f). To address whether FBXO7 directly interacts with PRMT1, the GST pulldown assay was performed using recombinant human GST-FBXO7 and His-MBP-PRMT1 proteins. As shown in Fig. 1g, a direct interaction between FBXO7 and PRMT1 was detected. By expressing different truncation mutants of GFP-PRMT1 and FLAG-FBXO7 and performing co-IP analysis, we found that the catalytic domain (23-162aa) of PRMT1 was required for its binding with FBXO7 (Fig. 1h–j), and the ubiquitin-like (UBL, 1-78aa) and FBXO7-PI31 dimerization (FP, 181-324aa) domains of FBXO7 were required for its binding with PRMT1 (Fig. 2k–m). Collectively, these data indicate that FBXO7 directly interacts with PRMT1.

### FBXO7 downregulates PRMT1 protein level by ubiquitin-mediated degradation in HCC cells

The interaction between the E3 ubiquitin ligase FBXO7 and PRMT1 leads us to presume that FBXO7 decreases PRMT1 protein level by ubiquitin-proteasome degradation. Indeed, *FBXO7* KD obviously augmented the protein level of PRMT1 (Supplementary Fig. 3a, b), whereas ectopic expression of GFP-FBXO7 markedly reduced PRMT1 protein level in a dose-dependent manner (Supplementary Fig. 3c). Interestingly, neither *FBXO7* KD (Supplementary Fig. 3d, e) nor FBXO7 over-expression (Supplementary Fig. 3f) affected the mRNA level of PRMT1 in Huh7 and PLC/PRF/5 cells. To determine whether FBXO7-mediated downregulation of PRMT1 protein level is attributed to ubiquitin-proteasome degradation, *FBXO7* KD HCC cells were treated with MG132 or bortezomib. Immunoblotting analysis using two different commercial PRMT1 antibodies demonstrated that the upregulated protein level of PRMT1 caused by *FBXO7* KD was not observed in MG132- or bortezomib-treated HCC cells (Fig. 2a, Supplementary Fig. 3g, h). In contrast, MG132 treatment significantly elevated PRMT1 protein level, and markedly abolished the inhibitory effect of GFP-FBXO7 on PRMT1 protein level (Fig. 2b). We then evaluated the protein stability of PRMT1 by measuring its protein half-life in Huh7 and PLC/PRF/5 cells treated with the protein synthesis inhibitor cycloheximide (CHX). Compared with control cells, *FBXO7* KD clearly prolonged the half-life of PRMT1 protein (Fig. 2c). Together, these data suggest that FBXO7 downregulates PRMT1 protein level probably due to ubiquitin-proteasome degradation.

To determine whether FBXO7 ubiquitinates PRMT1, we co-expressed GFP-PRMT1, wild-type (WT) or ΔF-box mutant (enzymatically dead mutant) of FLAG-FBXO7, and HA-ubiquitin (HA-Ub) in Huh7 cells. Expression of FLAG-FBXO7 WT, but not its enzymatically dead ΔF-box mutant, prominently increased the ubiquitination of GFP-PRMT1 in Huh7 cells (Fig. 2d, e). Consistently, the level of

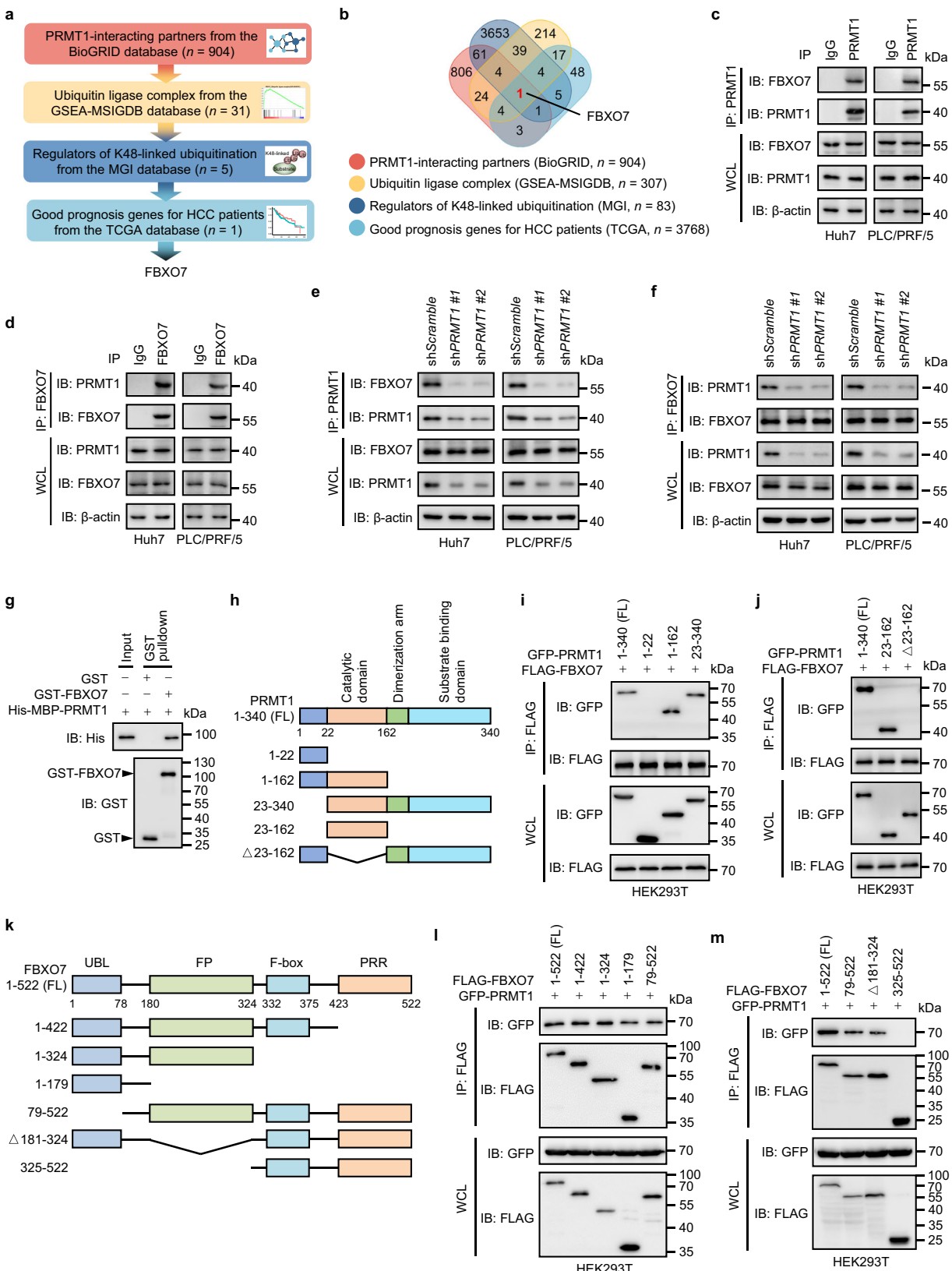

GFP-PRMT1 ubiquitination was largely attenuated by *FBXO7* KD in Huh7 and PLC/PRF/5 cells (Fig. 2f). Likewise, *FBXO7* KD led to a marked decrease of endogenous PRMT1 ubiquitination in Huh7 and PLC/PRF/5 cells (Fig. 2g). These results indicate that FBXO7 down-regulates PRMT1 through ubiquitin-mediated degradation in HCC cells.

**FBXO7 mediates PRMT1 ubiquitination at lysine 37 in HCC cells**

To identify the ubiquitination sites of PRMT1, ubiquitinated FLAG-PRMT1 was immunoprecipitated and subjected to LC-MS/MS analysis (Fig. 3a). 4 lysine residues (K37, K82, K202, K325) were identified as potential ubiquitination sites of PRMT1 (Fig. 3b). We then mutated these lysine residues to arginine (K37R, K82R, K202R, K325R) and

**Fig. 1 | FBXO7 interacts with PRMT1. a** Schematic of the screening workflow to identify the E3 ubiquitin ligase FBXO7 as a protein candidate downregulating PRMT1. **b** Venn diagram showing overlap of proteins among four datasets obtained from BioGRID, GSEA-MSIGDB, MGI, and TCGA. **c**, **d** Reciprocal co-IP analysis of PRMT1 and FBXO7 in Huh7 and PLC/PRF/5 cells. IgG was used as a negative control. The immunoblotting experiments were repeated three times with similar results. **e**, **f** Reciprocal co-IP analysis of PRMT1 and FBXO7 in *PRMT1* KD cells. The immunoblotting experiments were repeated three times with similar results. **g** Recombinant GST-FBXO7 protein was incubated with recombinant His-MBP-PRMT1 protein, followed by GST pulldown and immunoblotting analysis with GST and His antibodies. The immunoblotting experiments were repeated three times with similar results. **h** Schematic representation of full-length (FL) PRMT1 and

different truncation mutants. **i**, **j** GFP-PRMT1 FL or indicated truncation mutants were co-expressed with FLAG-FBXO7 in HEK293T cells. FLAG-FBXO7 was immunoprecipitated with FLAG beads, followed by immunoblotting analysis of GFP-PRMT1 using GFP antibody. The immunoblotting experiments were repeated three times with similar results. **k** Schematic representation of full-length (FL) FBXO7 and different truncation mutants. UBL ubiquitin-like domain, FP FBXO7-PI31 dimerization domain, PRR proline-rich region. **l**, **m** FLAG-FBXO7 FL or indicated truncation mutants were co-expressed with GFP-PRMT1 in HEK293T cells. FLAG-FBXO7 was immunoprecipitated with FLAG beads, followed by immunoblotting analysis of GFP-PRMT1 using GFP antibody. The immunoblotting experiments were repeated three times with similar results. Source data are provided as a Source Data file.

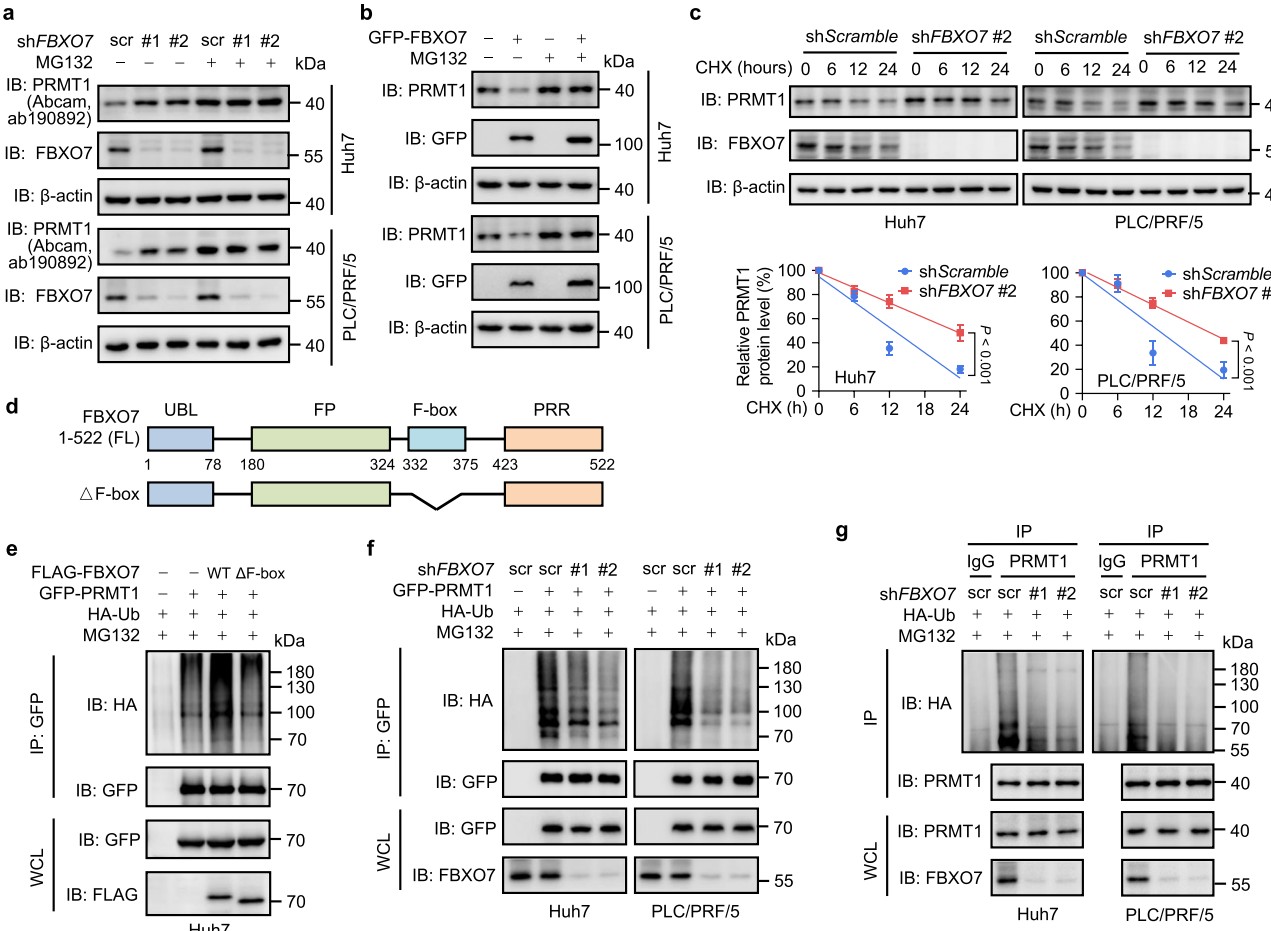

**Fig. 2 | FBXO7 promotes ubiquitin-mediated degradation of PRMT1 in HCC cells. a** Immunoblotting analysis of PRMT1 (anti-PRMT1 antibody: Abcam, ab190892) and FBXO7 in *FBXO7* KD cells treated with or without MG132 (25 μM, 6 h). The immunoblotting experiments were repeated three times with similar results. **b** Immunoblotting analysis of PRMT1 and GFP-FBXO7 in FBXO7-overexpressing cells treated with or without MG132 (25 μM) for 6 h. The immunoblotting experiments were repeated three times with similar results. **c** Immunoblotting analysis of PRMT1 and FBXO7 in *FBXO7* KD cells treated with or without cycloheximide (CHX, 50 μg/mL) for the indicated time. Quantitation of PRMT1 protein level based on band intensity was shown (bottom). Data are presented as the mean ± SD (*n* = 3 independent experiments with similar results). Statistical analysis was performed using the two-way ANOVA with Bonferroni correction. **d** Schematic representation of full-length (FL) FBXO7 and

enzymatically dead mutant (deletion of F-box domain, shown as ΔF-box). **e** GFP-PRMT1 and HA-ubiquitin (HA-Ub) were co-expressed with FLAG-FBXO7 WT or enzymatically dead ΔF-box mutant in Huh7 cells. After MG132 (25 μM, 6 h) treatment, IP was performed with GFP antibody, followed by immunoblotting with indicated antibodies. The immunoblotting experiments were repeated three times with similar results. **f** GFP-PRMT1 was co-expressed with HA-Ub in *FBXO7* KD cells. After MG132 (25 μM, 6 h) treatment, IP was performed with GFP antibody, followed by immunoblotting with indicated antibodies. The immunoblotting experiments were repeated three times with similar results. **g** *FBXO7* KD cells were transfected with HA-Ub plasmid and treated with MG132 (25 μM, 6 h). IP was performed with IgG or PRMT1 antibody, followed by immunoblotting with indicated antibodies. The immunoblotting experiments were repeated three times with similar results. Source data are provided as a Source Data file.

performed in vivo ubiquitination assay to examine which lysine is required for FBXO7-mediated PRMT1 ubiquitination. As shown in Fig. 3c, reconstituted expression of FLAG-FBXO7 in *FBXO7* KD Huh7 cells obviously elevated the ubiquitination level of wild-type GFP-PRMT1. This increase in PRMT1 ubiquitination could also be observed in K82R, K202R, and K325R mutants of GFP-PRMT1, but not in K37R mutant. Moreover, overexpression of FLAG-FBXO7 led to an apparent decrease in the level of wild-type PRMT1 but had no obvious effect on

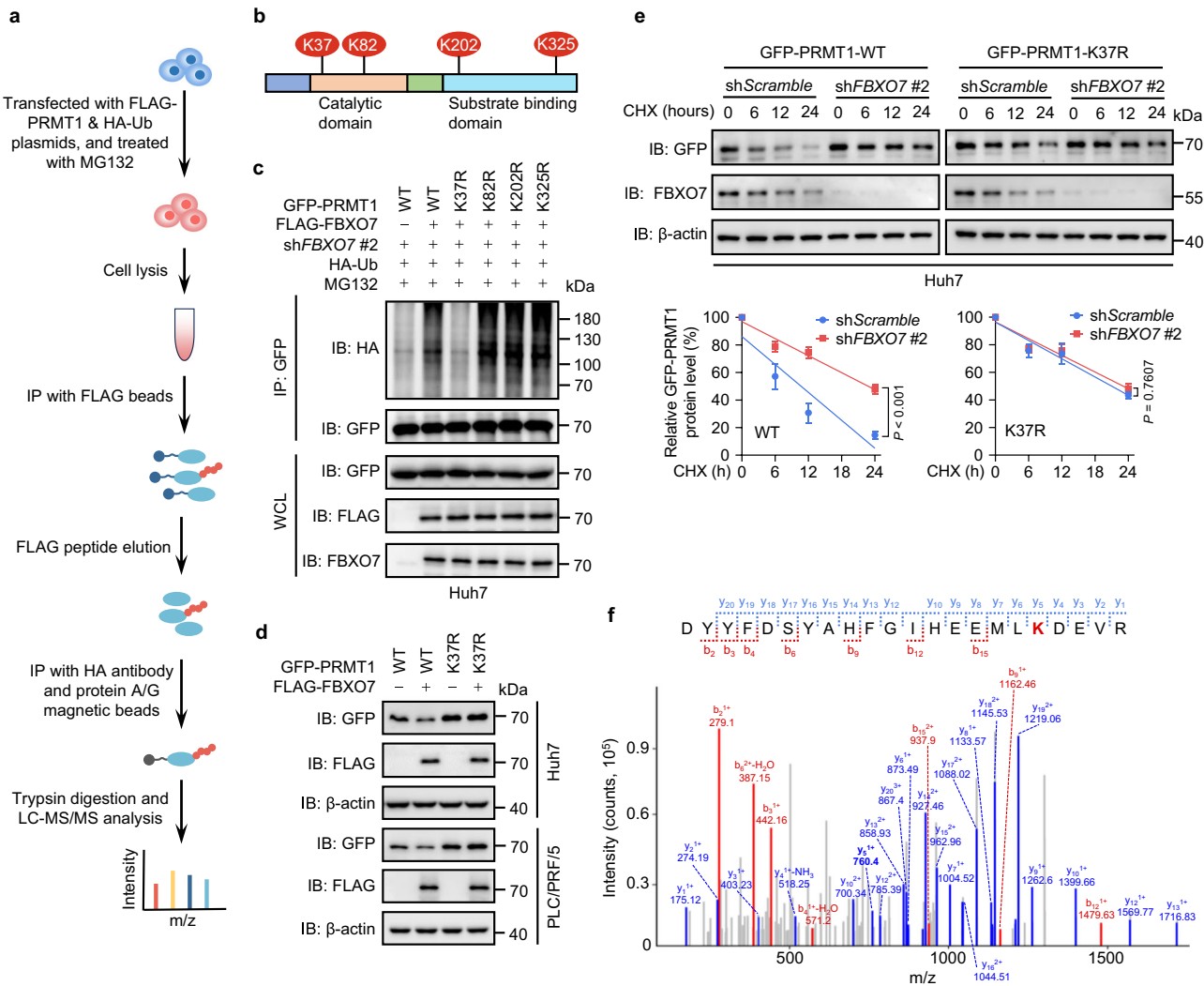

**Fig. 3 | FBXO7 mediates PRMT1 ubiquitination at lysine 37 in HCC cells.**
**a** Schematic of the workflow for identifying the ubiquitination sites of PRMT1 by mass spectrometry analysis. **b** Four potential lysine ubiquitination sites (K37, K82, K202, K325) in PRMT1 identified by mass spectrometry analysis. **c** GFP-PRMT1 (WT or indicated KR mutants) was co-expressed with HA-Ub in *FBXO7* KD Huh7 cells rescued with or without FLAG-FBXO7. After MG132 (25 μM, 6 h) treatment, IP was performed with GFP antibody, followed by immunoblotting with indicated antibodies. The immunoblotting experiments were repeated three times with similar results. **d** FLAG-FBXO7 was co-expressed with GFP-PRMT1 WT or K37R in Huh7 and PLC/PRF/5 cells, followed by immunoblotting with indicated antibodies. The

immunoblotting experiments were repeated three times with similar results. **e** *FBXO7* KD cells were transfected with GFP-PRMT1 WT or K37R plasmid, and treated with or without cycloheximide (CHX, 50 μg/mL) for the indicated time. Immunoblotting analysis of GFP-PRMT1 and FBXO7 was performed. Quantitation of GFP-PRMT1 protein level based on band intensity was shown (bottom). Data are presented as the mean ± SD ($n$ = 3 independent experiments with similar results). Statistical analysis was performed using the two-way ANOVA with Bonferroni correction. **f** Mass spectrometry identification of K37 ubiquitination of PRMT1. All immunoblotting experiments were repeated three times with similar results. Source data are provided as a Source Data file.

the K37R mutant (Fig. 3d). In addition, the K37R mutant of PRMT1 protein exhibited a longer half-life compared with wild-type PRMT1. Knockdown of *FBXO7* significantly increased the half-life of wild-type PRMT1 but had no obvious effect on that of the K37R mutant (Fig. 3e). The mass spectrometry identification of K37 ubiquitination for PRMT1 was also shown in Fig. 3f. Together, these data suggest that K37 is a major ubiquitination site of PRMT1 mediated by FBXO7.

**FBXO7 suppresses PHGDH methylation and activity by down-regulating PRMT1 in HCC cells.** Our previous study revealed that PHGDH is hyperactivated in HCC to promote serine synthesis and maintain redox homeostasis for HCC growth. This PHGDH hyper-activation is attributable to PRMT1 upregulation in HCC, which induces PHGDH methylation at arginine 236 and subsequent activation[15]. Given that FBXO7 downregulates PRMT1 by ubiquitin-mediated degradation, we determined whether FBXO7 reduces PHGDH methylation and

activity in HCC cells. Using a site-specific antibody recognizing mono-methylated R236 of PHGDH (mePHGDH (R236me1)) generated from our previous study[15], we found that *FBXO7* KD markedly enhanced PHGDH methylation level in Huh7 and PLC/PRF/5 cells, with a con-comitant increase in PRMT1 expression (Fig. 4a, Supplementary Fig. 4a). In contrast, enforced expression of FBXO7 led to obviously decreased PHGDH methylation level and reduced PRMT1 expression (Fig. 4b). To examine whether FBXO7 suppresses PHGDH R236 methylation through downregulating PRMT1, *PRMT1* was knocked down in *FBXO7* KD cells. As shown in Fig. 4c, the elevated R236 methylation level of PHGDH in *FBXO7* KD cells was prominently restored by *PRMT1* KD. Similarly, *PRMT1* KD failed to further decrease the reduced PHGDH methylation caused by FBXO7 overexpression (Supplementary Fig. 4b). Moreover, enforced expression of K37R mutant, but not wild-type PRMT1, obviously rescued the decreased level of PHGDH methylation in FBXO7-overexpressing cells (Fig. 4d).

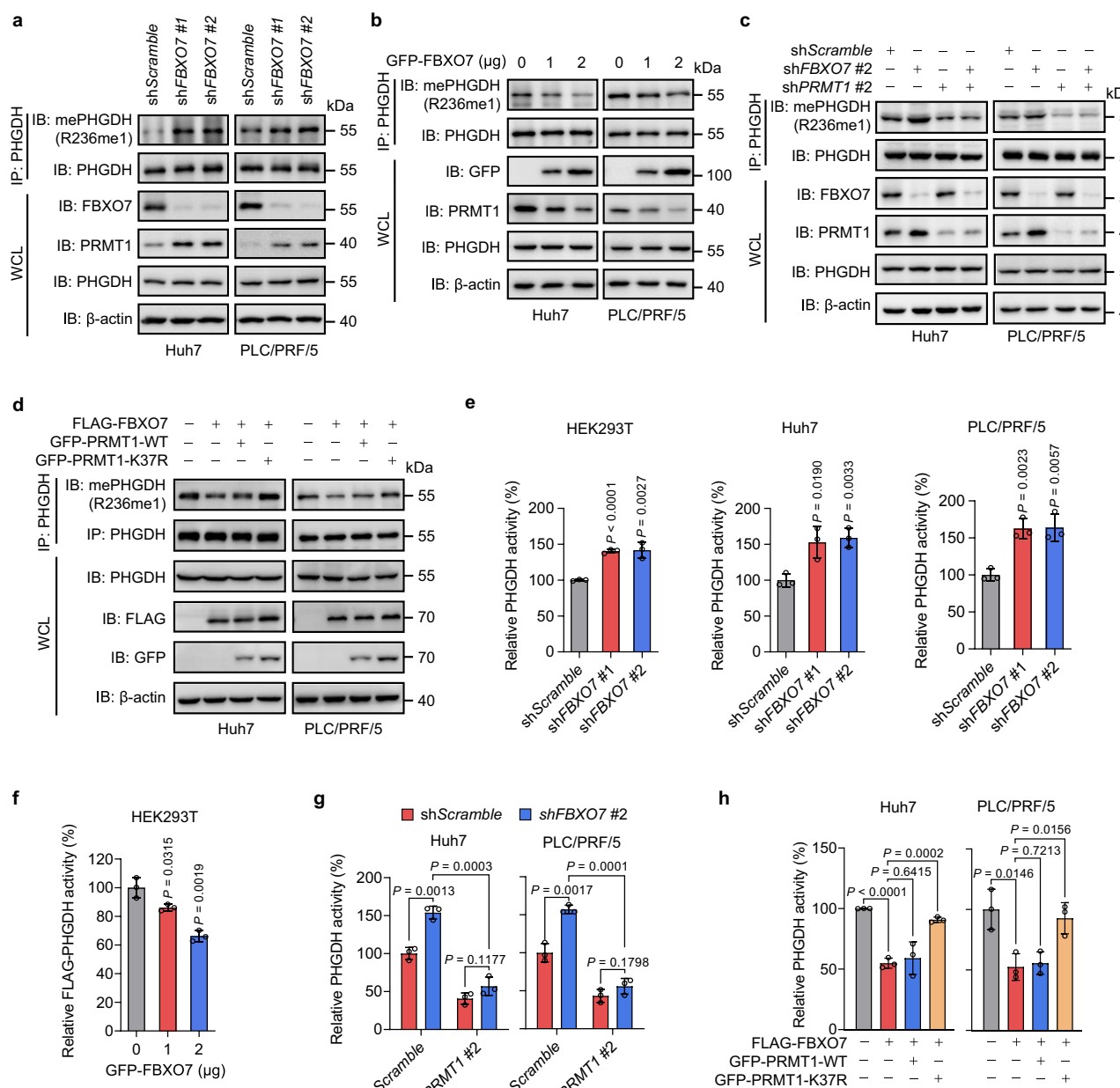

**Fig. 4 | FBXO7 suppresses PHGDH methylation and activity by downregulating PRMT1 in HCC cells. a, b** PHGDH was immunoprecipitated in *FBXO7* KD (**a**) or GFP-FBXO7-overexpressing (**b**) cells, followed by immunoblotting with indicated antibodies. The immunoblotting experiments were repeated three times with similar results. **c** PHGDH was immunoprecipitated in *FBXO7* and/or *PRMT1* KD cells, followed by immunoblotting with indicated antibodies. The immunoblotting experiments were repeated three times with similar results. **d** PHGDH was immunoprecipitated in cells overexpressing FLAG-FBXO7 and GFP-PRMT1-WT or K37R, followed by immunoblotting with indicated antibodies. The immunoblotting experiments were repeated three times with similar results. **e** Endogenous PHGDH was immunoprecipitated in HEK293T, Huh7, and PLC/PRF/5 cells with *FBXO7* KD, followed by measurement of PHGDH activity. Data are presented as the mean ± SD ($n$ = 3 independent experiments). Statistical analysis was performed using the two-

tailed Student's *t*-test. **f** GFP-FBXO7 plasmid was transfected in HEK293T cells stably expressing FLAG-PHGDH, followed by IP with FLAG beads and elution with FLAG peptides. PHGDH activity was then measured. Data are presented as the mean ± SD ($n$ = 3 independent experiments). Statistical analysis was performed using the two-tailed Student's *t*-test. **g** Endogenous PHGDH was immunoprecipitated in *FBXO7* and/or *PRMT1* KD cells, followed by measurement of PHGDH activity. Data are presented as the mean ± SD ($n$ = 3 independent experiments). Statistical analysis was performed using the two-tailed Student's *t*-test. **h** Endogenous PHGDH was immunoprecipitated in cells overexpressing FLAG-FBXO7 and GFP-PRMT1-WT or K37R, followed by measurement of PHGDH activity. Data are presented as the mean ± SD ($n$ = 3 independent experiments). Statistical analysis was performed using the two-tailed Student's *t*-test. Source data are provided as a Source Data file.

These data indicate that FBXO7 inhibits PHGDH methylation by downregulating PRMT1. *FBXO7* KD cells exhibited higher catalytic activity of PHGDH (Fig. 4e), while ectopic expression of FBXO7 inhibited PHGDH activity (Fig. 4f). In addition, the elevated PHGDH activity induced by *FBXO7* KD was markedly rescued by *PRMT1* KD (Fig. 4g). Consistently, FBXO7 overexpression caused reduced PHGDH activity

in control cells, but had no obvious effect in *PRMT1* KD cells (Supplementary Fig. 4c). The decreased PHGDH activity caused by FBXO7 overexpression was restored by K37R mutant of PRMT1, but not its wild-type (Fig. 4h). Overall, these results illustrate that FBXO7 suppresses PHGDH R236 methylation and catalytic activity by downregulating PRMT1 in HCC cells.

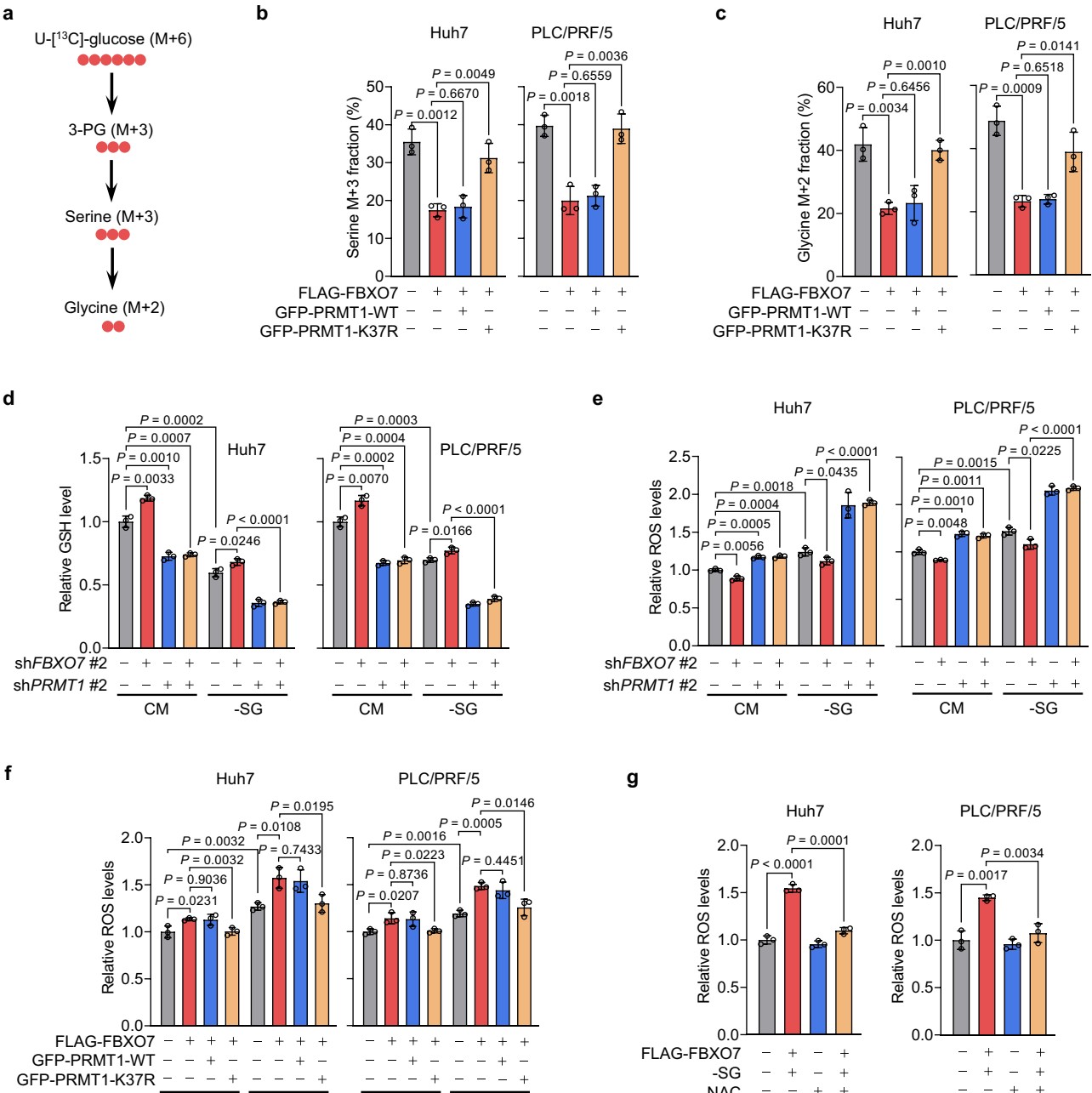

**Fig. 5 | FBXO7 inhibits serine synthesis and promotes oxidative stress by downregulating PRMT1 in HCC cells. a** Schematic of U-[13C]-glucose incorporation into serine and glycine in cells. **b, c** Incorporation of U-[13C]-glucose carbon into serine (**b**) and glycine (**c**) detected by LC–MS/MS in cells overexpressing FLAG-FBXO7 and GFP-PRMT1-WT or K37R. Data are presented as the mean ± SD (*n* = 3 independent experiments). Statistical analysis was performed using the two-tailed Student's *t*-test. **d, e** GSH (**d**) and ROS (**e**) levels in *FBXO7* and/or *PRMT1* KD cells cultured in CM or -SG medium. Data are presented as the mean ± SD (*n* = 3 independent experiments). Statistical analysis was performed using the two-tailed Student's *t*-test. **f** ROS levels in cells overexpressing FLAG-FBXO7 and GFP-PRMT1-WT or K37R cultured in CM or -SG medium. Data are presented as the mean ± SD (*n* = 3 independent experiments). Statistical analysis was performed using the two-tailed Student's *t*-test. **g** ROS levels in FLAG-FBXO7-overexpressing cells cultured in -SG medium in the presence or absence of NAC (5 mM). Data are presented as the mean ± SD (*n* = 3 independent experiments). Statistical analysis was performed using the two-tailed Student's *t*-test. Source data are provided as a Source Data file.

## FBXO7 inhibits serine synthesis and promotes oxidative stress by downregulating PRMT1 in HCC cells

PHGDH is a rate-limiting enzyme in the de novo serine synthesis pathway[3]. Given that FBXO7 downregulates PRMT1 and suppresses PHGDH activity, we next sought to assess whether FBXO7 influences serine synthesis by examining the total serine and glycine levels in HCC cells cultured with serine/glycine-free (-SG) medium. *FBXO7* KD in Huh7 and PLC/PRF/5 cells led to an obvious increase in both intracellular serine and glycine levels, which could be compromised by *PRMT1* KD (Supplementary Fig. 4d, e). In line with this, enforced expression of FBXO7 or *PRMT1* KD markedly reduced the levels of serine and glycine (Supplementary Fig. 4f, g). However, we could not observe a further decrease when FBXO7 was overexpressed in *PRMT1* KD cells (Supplementary Fig. 4f, g). To further confirm these observations, we used U-[13C]-glucose to trace the de novo synthesis of serine and glycine (Fig. 5a). As shown in Fig. 5b, c, the incorporation of

[13]C from glucose into serine and glycine was markedly diminished by enforced expression of FLAG-FBXO7, which could be rescued by overexpressing PRMT1 K37R mutant, but not PRMT1 WT. These data indicate that FBXO7 inhibits serine synthesis by down-regulating PRMT1.

Serine synthesis pathway plays a critical role in the maintenance of redox homeostasis by generating GSH and NADPH as its down-stream products[4,7]. To this end, we investigated whether FBXO7 is involved in regulating redox homeostasis by downregulating PRMT1. Consistent with our previous findings, serine/glycine depletion (−SG) led to obvious decreases in the GSH level, GSH/GSSG, and NADPH/NADP$^+$ ratios in Huh7 and PLC/PRF/5 cells (Fig. 5d, Supplementary Fig. 5a, b)[15]. The GSH level, GSH/GSSG, and NADPH/NADP$^+$ ratios were elevated in FBXO7 KD cells and were greatly restored by KD of PRMT1 in both serine/glycine repleted (complete medium; CM) and depleted (−SG) cells (Fig. 5d, Supplementary Fig. 5a, b). Moreover, FBXO7 overexpression caused reduced GSH levels, GSH/GSSG ratio, and NADPH/NADP$^+$ ratio in control cells, but not in PRMT1 KD cells (Supplementary Fig. 5c−e). The increased production of GSH and NADPH in FBXO7 KD cells resulted in a prominent reduction of reactive oxygen species (ROS) levels (Fig. 5e). This reduction of ROS levels caused by FBXO7 KD was significantly countered through PRMT1 KD (Fig. 5e). In support of these observations, ectopic FBXO7 expression promoted ROS accumulation in control cells, but had no obvious effect in PRMT1 KD cells (Supplementary Fig. 5f). Moreover, the increased ROS levels in FBXO7-overexpressing cells were significantly compromised by introducing K37R mutant, but not wild-type, of PRMT1 (Fig. 5f). In addition, ROS accumulation in FBXO7-overexpressing cells with ser-ine/glycine starvation was markedly attenuated by treatment of the antioxidant N-acetyl cysteine (NAC; Fig. 5g). Collectively, these data demonstrate that FBXO7 suppresses serine synthesis and induces oxidative stress by downregulating PRMT1 in HCC cells.

## FBXO7 suppresses HCC growth by downregulating PRMT1

Serine synthesis supports tumor cell growth by maintaining redox homeostasis, which is critical for the survival and growth of HCC cells[26,27]. Therefore, we further determined the role of FBXO7-mediated negative regulation of PRMT1 in HCC growth. Compared with control cells, the growth of PRMT1 KD cells was inhibited, with the growth inhibition effect being more prominent when serine and gly-cine in the medium were depleted (Fig. 6a). KD of FBXO7 led to an increase in the growth of control cells, but not in PRMT1 KD cells (Fig. 6a). In parallel, FBXO7 overexpression markedly suppressed the growth of both serine/glycine repleted and depleted HCC cells (Sup-plementary Fig. 6a). However, no obvious change in cell growth was observed when overexpressing FBXO7 in PRMT1 KD cells (Supple-mentary Fig. 6a). Interestingly, the suppressed growth of HCC cells caused by ectopic FBXO7 expression was significantly rescued by K37R mutant of PRMT1, but not wild-type PRMT1 (Fig. 6b). Moreover, FBXO7 overexpression led to cell cycle arrest (Supplementary Fig. 6b) and apoptosis induction (as evidenced by increased cleavage of caspase 3, Supplementary Fig. 6c) in HCC cells grown in serine/glycine-free (−SG) medium, which were prominently restored by K37R mutant of PRMT1, but not wild-type PRMT1. Notably, the decreased cell growth and increased caspase 3 cleavage in FBXO7-overexpressing cells with ser-ine/glycine starvation were markedly recovered by NAC treatment (Fig. 6c, d).

We then generated a mouse xenograft model by subcutaneously inoculating FBXO7 KD Huh7 cells with or without PRMT1 KD into nude mice fed with a control (+SG) or serine/glycine-free (−SG) diet. It was found that KD of PRMT1 suppressed the growth of tumor xenografts from mice fed with a +SG diet, as evidenced by decreased tumor size, volume, and weight. Notably, the suppression of tumor growth by PRMT1 KD was aggravated when mice were fed with a -SG diet (Fig. 6e, f, Supplementary Fig. 6d). FBXO7 KD led to an obvious increase in tumor growth. However, this increase was profoundly abolished by KD of PRMT1 (Fig. 6e, f, Supplementary Fig. 6d). A similar trend for the proliferation of tumor cells was also observed as reflected by Ki-67 staining (Supplementary Fig. 6e, f). Moreover, immunohisto-chemical (IHC) staining revealed that tumor tissues with FBXO7 KD exhibited increased levels of PRMT1 protein and PHGDH R236 methylation, which was largely restored by PRMT1 KD (Supplementary Fig. 6e). To further support these observations, we also sub-cutaneously injected Huh7 cells overexpressing FLAG-FBXO7 and GFP-PRMT1-WT or K37R into nude mice. As shown in Fig. 6g, h, over-expression of FLAG-FBXO7 inhibited the growth of xenografts from mice fed with a +SG diet. A further decrease in tumor size and weight was observed when mice were fed with a −SG diet (Fig. 6g, h). The inhibition effect of FLAG-FBXO7 on tumor growth was markedly recovered by the K37R mutant of PRMT1, but not wild-type PRMT1 (Fig. 6g, h). Notably, tumor xenograft with FLAG-FBXO7 over-expression showed the decreased intensity of Ki-67 staining, accom-panied by increased intensity of 8-oxo-dG (a marker of oxidative stress) and cleaved caspase 3 (an apoptosis marker) staining, all of which were obviously rescued by expressing PRMT1 K37R mutant, but not PRMT1 WT (Fig. 6i, Supplementary Fig. 6g). Taken together, these results suggest FBXO7 suppresses PHGDH methylation and HCC growth by downregulating PRMT1.

## FBXO7 is negatively correlated with PRMT1 and PHGDH arginine methylation in HCC tissues

Next, we investigated the clinical relevance of FBXO7-mediated inhi-bition of PRMT1 expression and PHGDH methylation in HCC. IHC analysis of 45 paired human HCC samples showed that FBXO7 protein level was downregulated in HCC tissues relative to normal adjacent tissues (Fig. 7a, b), while PRMT1 protein level and PHGDH R236 mono-methylation level were both upregulated (Fig. 7c−f). Notably, FBXO7 protein level was inversely correlated with the level of PRMT1 protein and PHGDH R236 methylation, respectively (Fig. 7g−j). Reciprocal co-IP analysis showed that FBXO7 interacted with PRMT1 in six human HCC tissues, and this interaction level was positively correlated with the ubiquitination level of PRMT1 (Fig. 7k, l). Moreover, we analyzed the survival rate of HCC patients using the KM plotter database and found that a higher FBXO7 level was correlated with longer overall and progression-free survival (Supplementary Fig. 7a, b). These data, together with our findings in HCC cells and mouse xenograft model, suggest that downregulation of FBXO7 in human HCC is a major mechanism contributing to PRMT1-mediated elevation of PHGDH R236 mono-methylation and activity in HCC. The tumor suppressor gene TP53 is mutated in 31% of HCC patients, and p53 target gene expression signature (p53 signature) correlating with patient survival was recently developed[28]. We thus stratified 186 HCC patients in TCGA database by p53 signature, showing that patients with low p53 signature exhibited reduced overall survival (Supplementary Fig. 7c). Interestingly, p53 signature was positively correlated with the mRNA level of FBXO7 (Supplementary Fig. 7d), implying that ther-apeutic strategies targeting FBXO7−PRMT1−PHGDH axis might be applicable for HCC patients with low p53 signature, which requires further investigation.

## Discussion

In this study, we show a tumor-suppressive role of FBXO7 by inhibiting serine synthesis in HCC. As an E3 ubiquitin ligase, FBXO7 directly interacts with PRMT1, leading to PRMT1 ubiquitination and sub-sequent ubiquitin-proteasome degradation. FBXO7-mediated degra-dation of PRMT1 prevents PHGDH methylation and activation, thereby inhibiting serine synthesis, aggravating oxidative stress, and suppres-sing HCC growth in vitro and in vivo (Fig. 7m). Our study, therefore, unravels FBXO7−PRMT1−PHGDH axis a critical mechanism underlying serine metabolism regulation in HCC.

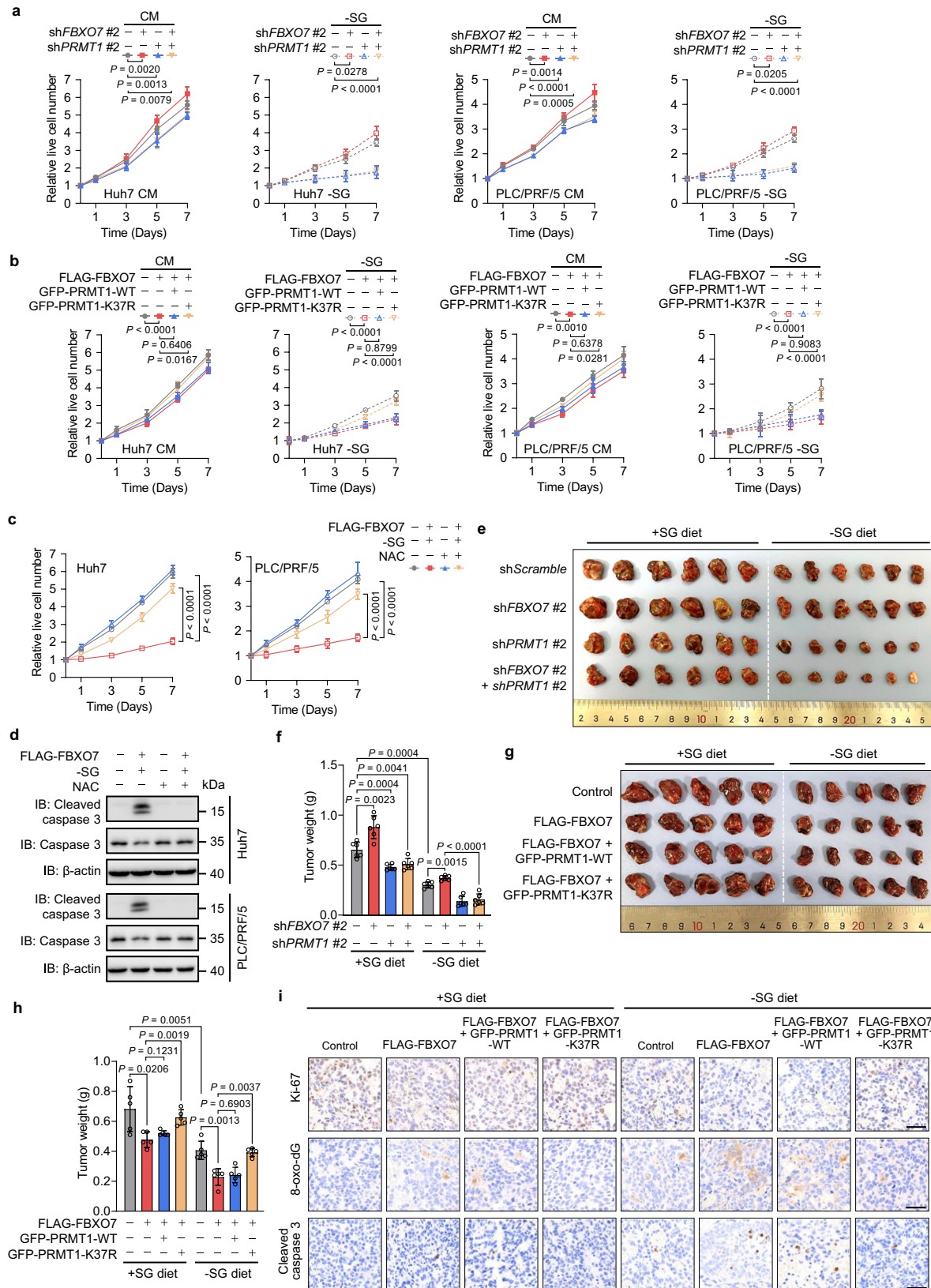

The E3 ligase FBXO7 has both ubiquitin-dependent and independent functions and is closely associated with Parkinson's disease and cancer[16]. It has been reported that FBXO7 is highly expressed in human lung cancer and colon cancer, suggesting a tumor-promoting role of FBXO7. Indeed, FBXO7 can bind CDK6 and stimulate cyclin D/CDK6 activity, leading to fibroblast transformation and tumorigenesis in

mice[19]. In addition, FBXO7 maintains mesenchymal and immune evasion phenotypes of cancer cells by counteracting FBXW7-mediated proteasomal degradation of eyes absent homolog 2 (EYA2)[20]. However, recent studies also reveal a tumor-suppressive role of FBXO7. For example, loss of FBXO7 induces chromosome instability and contributes to tumorigenesis in colorectal cancer[29]. Moreover, FBXO7 is

**Fig. 6 | FBXO7 suppresses HCC growth by downregulating PRMT1. a** Growth rates of *FBXO7* and/or *PRMT1* KD cells grown in CM or −SG medium. Data are presented as the mean ± SD (*n* = 5 independent experiments). Statistical analysis was performed using the two-way ANOVA with Bonferroni correction. **b** Growth rates of cells overexpressing FLAG-FBXO7 and GFP-PRMT1-WT or K37R grown in CM or −SG medium. Data are presented as the mean ± SD (*n* = 5 independent experiments). Statistical analysis was performed using the two-way ANOVA with Bonferroni correction. **c** Growth rates of FLAG-FBXO7-overexpressing cells grown in -SG medium in the presence or absence of NAC (5 mM). Data are presented as the mean ± SD (*n* = 5 independent experiments). Statistical analysis was performed using the two-way ANOVA with Bonferroni correction. **d** Immunoblotting analysis of cleaved caspase 3 and caspase 3 in FLAG-FBXO7-overexpressing cells grown in −SG medium in the presence or absence of NAC (5 mM). The immunoblotting experiments were repeated three times with similar results. **e, f** *FBXO7* and/or *PRMT1* KD Huh7 cells were subcutaneously inoculated into nude mice fed with a +SG or −SG diet. Tumor image (**e**) and weight (**f**) were shown. Data are presented as the mean ± SD (*n* = 6 mice). Statistical analysis in **f** was performed using the two-tailed Student's *t*-test. **g, h** Huh7 cells overexpressing FLAG-FBXO7 and GFP-PRMT1-WT or K37R were subcutaneously inoculated into nude mice fed with a +SG or −SG diet. Tumor image (**g**) and weight (**h**) were shown. Data are presented as the mean ± SD (*n* = 5 mice). Statistical analysis in **h** was performed using the two-tailed Student's *t*-test. **i** Representative images of IHC staining for Ki-67, 8-oxo-dG, and cleaved caspase 3 in tumor xenografts in (**g**). Scale bars, 50 µm. Source data are provided as a Source Data file.

downregulated in endometrial carcinoma. FBXO7 ubiquitinates inverted formin 2 (INF2) for degradation, leading to the suppression of mitochondrial division and tumor progression in endometrial carcinoma[24]. In our study, we found that FBXO7 is downregulated in HCC and acts as a tumor suppressor in HCC by ubiquitinating and degrading PRMT1, which is consistent with a previous report showing that FBXO7 induces ubiquitination and degradation of hepatoma up-regulated protein (HURP)[30]. Notably, a very small portion of FBXO7-overexpressing HCC cells still survived when serine and glycine were deprived, which might be attributed to the activation of salvage pathways for GSH and NADPH synthesis or GSH- and NADPH-independent antioxidant pathways. It seems that the reported onco-genic role of FBXO7 is related to its E3 ligase-independent function, while both the E3 ligase-dependent and independent functions of FBXO7 are associated with its tumor-suppressive function. Therefore, FBXO7 has a context-dependent dual role in tumor progression, which merits further investigation.

Compared with its atypical function, the E3 ligase function of FBXO7 is poorly defined, especially in cancer biology. Currently, only a handful of substrates have been identified for FBXO7, such as HURP, phosphofructokinase (PFKP), FOXO4, SIRT7, and INF2[23,24,30–32]. Here, we identified PRMT1 as an unreported substrate for FBXO7 in HCC. It has been reported that PRMT1 is frequently upregulated in HCC, promoting HCC growth and metastasis[13–15,25]. However, the mechanism underlying aberrant PRMT1 upregulation in HCC is poorly understood. We found that FBXO7 is downregulated in HCC, and the decrease in FBXO7-mediated K37 ubiquitination and proteasomal degradation of PRMT1 contributes to the elevated PRMT1 protein level in HCC. However, there is no HCC-associated K37 mutation of PRMT1 resisting FBXO7-mediated ubiquitination in the COSMIC database. This might be due to the possibility that FBXO7 is frequently downregulated in HCC patients by the mutation of *TP53* or other tumor suppressor genes, leading to high PRMT1 protein abundance to support HCC development.

Serine metabolism supports biomass, energy, and reductant demands for cancer cell survival and proliferation[4]. Targeting serine metabolism has revealed favorable anticancer effects in various pre-clinical studies[9]. As a major source of intracellular serine pool, the serine synthesis pathway has attracted much attention for its down-stream epigenetic regulation through S-adenosyl methionine (SAM) production and redox balance maintenance through GSH and NADPH generation[8]. However, how serine synthesis is regulated in cancer remains largely to be defined. Our previous study found that PHGDH is hyperactivated to promote serine synthesis in HCC, which is caused by PRMT1 upregulation and PRMT1-mediated PHGDH mono-methylation at arginine 236[15]. In this study, we identified the downregulation of FBXO7 as a major cause of PRMT1 upregulation in HCC. Therefore, our study demonstrated that the FBXO7–PRMT1–PHGDH axis plays a crucial role in regulating serine metabolism in HCC, and will provide insights for the development of serine-targeting strategies in cancer therapy.

# Methods

## Study approval

The HCC tissues used in this study were obtained from West China Hospital, Chengdu, with informed written consent from patients. The use of human specimens was approved by the Institutional Ethics Committee of Sichuan University. There was no bias in the selection of patients. All animal studies were performed in accordance with guidelines provided by the Institutional Animal Care and Treatment Committee of Sichuan University. The animals were treated in accordance with relevant institutional and national guidelines and regulations. The maximal allowable tumor size/burden (diameter ≤ 1.5 cm) was not exceeded. Sex/gender was not considered in the study design and analysis.

## Cell culture

Huh7 (TCHu182), PLC/PRF/5 (TCHu119), and HEK293T (GNHu17) cells were obtained from the Bank of Type Culture Collection of the Chinese Academy of Sciences. All cells were cultured in Dulbecco's Modified Eagle Medium (DMEM) supplemented with 10% fetal bovine serum (FBS; VivaCell Biosciences, C04001) and 1% penicillin-streptomycin in a humidified chamber at 37 °C under 5% $CO_2$ atmosphere. All cells were mycoplasma-free and authenticated by fingerprinting of short tandem repeats. For serine and glycine starvation, cells were maintained in serine- and glycine-free DMEM (US Biological Life Sciences, D9800-03) supplemented with 10% dialyzed FBS (VivaCell Biosciences, C3820).

## Reagents and antibodies

L-serine (S4311), glycine (G8790), NAD+ (N1511), 3-phosphoglycerate (3-PG; P8877), glutamate (G1626), and N-acetyl cysteine (NAC; A9165) were purchased from Sigma-Aldrich. MG132 (HY-13259), CHX (HY-12320), and bortezomib (HY-10227) were purchased from MedChem-Express. Lipofectamine 3000 (L3000015) was purchased from Thermo Fisher Scientific. U-[13C]-glucose (CLM1396) was purchased from Cambridge Isotope Laboratories.

Antibodies against PHGDH (ab57030, 1:1000), GFP (ab32146, 1:5000), PRMT1 (ab190892, 1:1000 for immunoblotting, and 1:100 for IHC staining), HA (ab18181, 1:4000; ab9110, 1:5000), and 8-oxo-dG (ab206461, 1:50 for IHC staining) were purchased from Abcam. Anti-bodies against PHGDH (PA5-27578, 1:1000) and FBXO7 (PA5-115219, 1:500 for immunoblotting, and 1:100 for IHC staining) were purchased from Thermo Fisher Scientific. Antibody against cleaved caspase 3 (9661, 1:500 for immunoblotting, and 1:200 for IHC staining) was purchased from Cell Signaling Technology. Antibody against GFP (11814460001, 1:2000) was purchased from Roche. Antibodies against FLAG (F3165, 1:4000 for immunoblotting; F1804, 1:100 for immuno-fluorescence), and Ki-67 (AB9260, 1:200 for IHC staining) were pur-chased from Sigma-Aldrich. Antibody against FBXO7 (sc-271763, 1:500) was purchased from Santa Cruz Biotechnology. Antibodies against GST (10000-0-AP, 1:2000), and His (66005-1-Ig, 1:4000) were pur-chased from Proteintech. Antibodies against caspase 3 (A0214, 1:1000), and β-actin (AC026, 1:4000) were purchased from ABclonal.

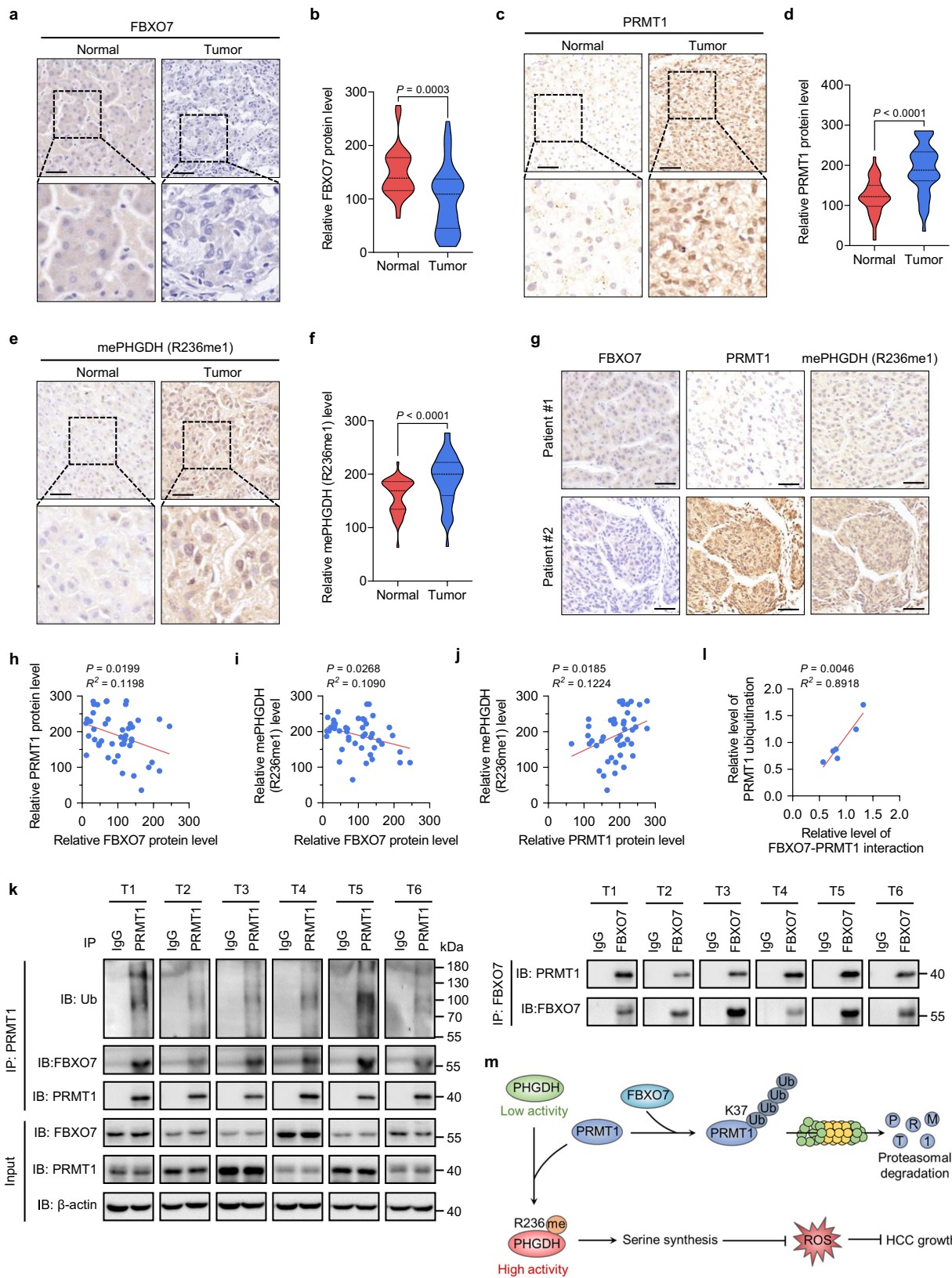

Antibody against ubiquitin (PTM-1106RM, 1:1000) was purchased from PTM BIO. The site-specific antibody recognizing mono-methylated R236 of PHGDH (mePHGDH (R236me1), 1:500 for immunoblotting, and 1:50 for IHC staining) was generated by immunizing rabbits with the synthesized peptide RVVNAA-R(me1)-GGIVDC (GL Biotech, Shanghai, China).

siRNA sequences (synthesized by GenePharma) are listed as follows:

si*PRMT1* #1: 5′-CGUCAAAGCCAACAAGUUA-3′
si*PRMT1* #2: 5′-GGACAUGACAUCCAAAGAU-3′
si*FBXO7* #1: 5′-CUGAGUCAAUUCAAGAUAA-3′
si*FBXO7* #2: 5′-ACAUCUGGUAGGUUCAGUG-3′

**Fig. 7 | FBXO7 is negatively correlated with PRMT1 and PHGDH arginine methylation in HCC tissues. a**, **b** Representative images (**a**) and quantitative analysis (**b**) of IHC staining for FBXO7 in HCC tissues and paired normal tissues ($n = 45$ samples). Scale bars, 50 μm. Statistical analysis was performed using the paired two-tailed Student's *t*-test. **c**, **d** Representative images (**c**) and quantitative analysis (**d**) of IHC staining for PRMT1 in HCC tissues and paired normal tissues ($n = 45$ samples). Scale bars, 50 μm. Statistical analysis was performed using the paired two-tailed Student's *t*-test. **e** and **f** Representative images (**e**) and quantitative analysis (**f**) of IHC staining for mePHGDH (R236me1) in HCC tissues and paired normal tissues ($n = 45$ samples). Scale bars, 50 μm. Statistical analysis was performed using the paired two-tailed Student's *t*-test. **g** Representative images of IHC staining with FBXO7, PRMT1, and mePHGDH (R236me1) antibodies in HCC tissues ($n = 45$ samples). Scale bars, 50 μm. **h** Two-tailed Pearson correlation test analyzing the relationship between the IHC staining intensity of FBXO7 and PRMT1 in HCC tissues ($n = 45$ samples). **i** Two-tailed Pearson correlation test analyzing the relationship between the IHC staining intensity of FBXO7 and mePHGDH (R236me1) in HCC tissues ($n = 45$ samples). **j** Two-tailed Pearson correlation test analyzing the relationship between the IHC staining intensity of PRMT1 and mePHGDH (R236me1) in HCC tissues ($n = 45$ samples). **k** IP and immunoblotting analysis with indicated antibodies in 6 HCC tissues (T1–T6). The immunoblotting experiments were repeated three times with similar results. **l** Two-tailed Pearson correlation test analyzing the relationship between the levels of FBXO7–PRMT1 interaction and PRMT ubiquitination in HCC tissues based on band intensity in **k** ($n = 6$ samples). **m** Schematic model for the mechanism of FBXO7–PRMT1–PHGDH axis in regulating serine synthesis, oxidative stress, and HCC growth. Source data are provided as a Source Data file.

## DNA constructs and mutagenesis

The PCR-amplified human FBXO7 cDNA was inserted into the pEGFP-N1 or pCDH-FLAG-puro vector. GFP-PRMT1 plasmid was a gift from Dr. Qunying Lei and Dr. Yiping Wang (Fudan University, China), which was originally provided by Dr. Yanzhong (Frankie) Yang from the City of Hope Medical Center. The point mutations of PRMT1 (K37R, K82R, K202R, K325R) were then generated using a Fast Site-directed Mutagenesis Kit (TransGen Biotech, FM111). pLenti-6.3-PHGDH-FLAG plasmid was obtained from our previous study[15].

## Generation of stable cell pools

To generate *FBXO7* or *PRMT1* stable knockdown (KD) cell pools, shRNA sequences targeting *FBXO7* or *PRMT1* were cloned into pMKO.1-puro vector. The retrovirus was produced by using a two-plasmid packaging system. Cells were infected with the retrovirus and selected with puromycin (2 μg/mL) for 1 week. The shRNA sequences were as follows:

sh*FBXO7*#1 (targeting CDS): 5'-GCCACATTCATTAGAGACCTT-3'
sh*FBXO7*#2 (targeting 3'UTR): 5'-CCCTGCTCTTGGTTCTCCTCT-3'
sh*PRMT1*#1 (targeting CDS): 5'-TTGACTCCTACGCACACTTTG-3'
sh*PRMT1*#2 (targeting 3'UTR): 5'-TGAGCGTTCCTAGGCGGTT TC-3'

pCDH-FLAG vector and pLVX-AcGFP1-C1 vector were used to stably express FBXO7 and PRMT1 in HCC cells, respectively. The lentivirus was produced using a two-plasmid packaging system and transfected in cells, followed by selection with puromycin (2 μg/mL) for 1 week.

## Immunoblotting and immunoprecipitation

Immunoblotting analysis was performed[33]. In brief, protein lysates from cultured cells were lysed with cold RIPA lysis buffer (50 mM Tris (pH 7.4), 150 mM NaCl, 1.0 mM EDTA, 1% NP-40, 1% sodium deoxycholate) containing protease inhibitor cocktail (Bimake, B14002) and phosphatase inhibitors (Bimake, B15002). The lysates were quantitated with Bradford reagent, separated by SDS-PAGE, transferred to PVDF membranes, and then incubated with primary and secondary antibodies. The blots were visualized by Immobilon Western HRP Substrate (EMD Millipore, WBKLS0500), and the images were captured using a ChemiScope 6000 Touch (Clinx Science Instruments).

For immunoprecipitation, NP-40 buffer (50 mM Tris–HCl (pH 7.4), 150 mM NaCl, 1% NP-40, 1.0 mM EDTA) containing protease inhibitor cocktail and phosphatase inhibitors (Bimake) was used for cell lysis. Protein lysates were incubated with FLAG beads (Sigma-Aldrich, A2220), or protein A/G magnetic beads (MCE, HY-K0202) conjugated with primary antibody at 4 °C for 3 h. The beads were then washed with NP-40 buffer and analyzed by immunoblotting protocol.

## PHGDH activity assay

The enzymatic activity of PHGDH was measured according to the previous report[15]. In detail, PHGDH-FLAG protein was immunopurified with FLAG beads (Sigma-Aldrich, A2220) followed by elution with FLAG peptides (Sigma-Aldrich, F4799). PHGDH-FLAG protein and recombinant PSAT1 protein (to prevent the product inhibition of PHGDH) were then added to the assay buffer (50 mM Tris–HCl, pH 8.5, 1 mM EDTA, 120 μM NAD$^+$, 240 μM 3-PG, and 30 mM glutamate) in 96-well plates. PHGDH activity was examined at 28 °C by measuring the NADH fluorescence (Ex/Em = 340/460 nm) using a Varioskan LUX Multimode Microplate Reader (Thermo Fisher Scientific). To measure PHGDH activity in HCC tissues, PHGDH protein was immunoprecipitated from tissue lysates with protein A/G magnetic beads (MCE, HY-K0202) conjugated with PHGDH antibody. The beads were then suspended in an assay buffer for the measurement of NADH fluorescence.

## Measurement of GSH level, NADPH/NADP$^+$ ratio, and ROS levels

The GSH level and GSH/GSSG ratio were measured using GSH and GSSG Assay Kit (Beyotime, S0053) following the manufacturer's instructions. GSH and GSSG signal intensities were measured at OD 412 nm using a Varioskan LUX Multimode Microplate Reader (Thermo Fisher Scientific). The NADPH/NADP$^+$ ratio was determined using the NADP$^+$/NADPH Assay Kit (Beyotime, S0179) according to the manufacturer's instructions. NADPH and NADP$^+$ signal intensities were measured at OD 450 nm using a Varioskan LUX Multimode Microplate Reader (Thermo Fisher Scientific). For the measurement of ROS levels, cells were washed with cold PBS twice and then stained with the fluorescent dye 2′,7′-dichlorofluorescein diacetate (H$_2$DCF-DA; Sigma-Aldrich, 35845) for 30 min at 37 °C. The stained cells were washed, trypsinized, and resuspended in PBS. A Varioskan LUX Multimode Microplate Reader (Thermo Fisher Scientific) was used to measure the fluorescent intensity (Ex/Em = 488/525 nm).

## GST pulldown assay

Recombinant human GST-FBXO7 protein (Novus Biologicals, H00025793-P01) or GST protein was mixed with recombinant human His-MBP-PRMT1 protein (Novus Biologicals, NBC1-18446) in NP-40 buffer at 4 °C overnight. The protein mixture was then incubated with GST beads (Smart Lifesciences, SA008005) at 4 °C for 3 h. After washing with NP-40 buffer, Beads were boiled with SDS loading buffer for immunoblotting analysis.

## Immunofluorescence

Huh7 and PLC/PRF/5 cells expressing FLAG-FBXO7 and GFP-PRMT1 were seeded on glass coverslips in 24-well plates, washed with PBS, and then fixed by 4% paraformaldehyde fixation. Cells were subsequently incubated with PBS containing 0.2% Triton X-100 and 5% BSA for 1 h, followed by incubation with FLAG antibody at 4 °C overnight and goat anti-mouse Alexa Fluor 594 (Thermo Fisher Scientific, A32742) at room temperature for 1.5 h. Images were captured using a Zeiss LSM 710 confocal microscope.

## Protein half-life assay

To determine the protein half-life of PRMT1, cells were treated with CHX (50 μg/mL) for the indicated time periods (0, 6, 12, and 24 h), followed by immunoblotting analysis.

## Identification of ubiquitination sites by mass spectrometry analysis

The ubiquitination sites of PRMT1 were identified by mass spectrometry analysis ($n = 1$). Briefly, HEK293T cells stably expressing FLAG-PRMT1 were transfected with HA-ubiquitin (HA-Ub) plasmids followed by MG132 ($25\,\mu M$) treatment for 6 h. Cells were then lysed and subjected to immunoprecipitation using FLAG beads (Sigma-Aldrich, A2220). After washing with NP-40 buffer, FLAG-PRMT1 protein was eluted from the immunoprecipitates by FLAG peptides (Sigma-Aldrich, F4799). The eluted FLAG-PRMT1 protein was then incubated with HA antibody at 4 °C overnight, followed by incubation with protein A/G magnetic beads (MCE, HY-K0202) for 3 h. The beads conjugating with ubiquitinated PRMT1 protein were then digested with sequencing-grade trypsin (Promega, V5111), and subjected to NSI source followed by tandem mass spectrometry (MS/MS) analysis in Q Exactive™ Plus (Thermo Fisher Scientific) coupled online to the UPLC (PTM BIO). The $m/z$ scan range was 350–1800 for a full scan, and intact peptides were detected in the Orbitrap at a resolution of 70,000. Peptides were then selected for MS/MS using NCE setting as 28, and the fragments were detected in the Orbitrap at a resolution of 17,500. A data-dependent procedure that alternated between one MS scan followed by 20 MS/MS scans with 15.0 s dynamic exclusion. Automatic gain control (AGC) was set at 5E4. The MS/MS data were processed using Proteome Discoverer 1.3 software (Thermo Fisher Scientific).

## Label-free quantitative proteomics analysis

In brief, Huh7 cells stably expressing FLAG-PRMT1 (cells expressing FLAG vector as control group) were lysed and immunoprecipitated by FLAG beads ($n = 2$). Immunoprecipitates were then digested with sequencing-grade trypsin and desalted. The desalted peptides were separated on an EASY-nLC 1200 UPLC system (Thermo Fisher Scientific) and then subjected to MS/MS analysis with Orbitrap Exploris 480 (Thermo Fisher Scientific). The full MS scan resolution was set to 60,000 for a scan range of 400–1200 $m/z$. The MS/MS scan was fixed first mass as 110 $m/z$ at a resolution of 30,000, with the TurboTMT set as off. Up to 15 most abundant precursors were then selected for further MS/MS analyses with 30 s dynamic exclusion. The HCD fragmentation was performed at a normalized collision energy (NCE) of 27%. The automatic gain control (AGC) target was set at 75%, with an intensity threshold of 10000 ions/s and a maximum injection time of 100 ms. The resulting MS/MS data were processed using Proteome Discoverer search engine (v.2.4) for label-free quantitative analysis.

## Measurement of serine and glycine levels

Cells were cultured in six-well plates with serine- and glycine-free DMEM supplemented with 10% dialyzed FBS. Serine and glycine concentrations in cell lysates were then determined using DL-Serine Assay Kit (Biovision, K743) and Glycine Assay Kit (Biovision, K589) respectively, following the manufacturer's instructions. A Varioskan LUX Multimode Microplate Reader (Thermo Fisher Scientific) was used to measure the fluorescent intensity (Ex/Em = 535/587 nm).

## Metabolic flux analysis with ¹³C isotopic tracers

$5 \times 10^6$ Huh7 cells were washed with PBS twice and then cultured in a glucose-free medium supplemented with 10% dialyzed FBS and 25 mM U-[$^{13}$C]-glucose for 24 h. Cells were then washed twice with cold PBS, followed by resuspension with pre-cold 80% methanol and 20% ddH$_2$O for metabolite extraction. LC–MS/MS analysis of the abundance serine and glycine with U-[$^{13}$C]-glucose carbon incorporation was conducted by LipidALL Technologies (Changzhou, China).

## Quantitative RT-PCR analysis

Quantitative RT-PCR analysis was performed[34]. Briefly, Total RNA was extracted using TRIzol (Thermo Fisher Scientific, 15596026), and was reverse-transcribed using the PrimeScript™ RT reagent Kit with gDNA

Eraser (Takara, RR047A). The mRNA level of *PRMT1* was quantitated using the Bio-Rad iTaq Universal SYBR Green Supermix (Bio-Rad, 1725271) using a CFX96 Connect Real-Time System (Bio-Rad Laboratories). Primer pairs used for *PRMT1* were: forward 5′-CCATTT-GAAAGTGGGAAACCA-3′ and reverse 5′-CATCATTCACTGCAACCTT GAAG-3′.

## Animal studies

BALB/c nude mice (6-week-old male, HFK Bioscience) were housed under ambient temperature of $24 \pm 2$ °C, circulating air, constant humidity of $50 \pm 10\%$, and a 12 h:12 h light/dark cycle. $5 \times 10^6$ Huh7 cells were suspended in PBS and inoculated into the flanks of nude mice. For serine and glycine starvation, mice were fed with the serine- and glycine-free (−SG) diet (HFK Bioscience) 5 days after tumor injection. Tumor volume was measured every other day. Mice were euthanized by cervical dislocation at the end of the experiment, after which xenografts were collected for further analysis.

## Immunohistochemical (IHC) staining

The tumor xenografts were fixed in 4% paraformaldehyde followed by paraffin embedding, and then analyzed by IHC staining to measure the protein levels of PRMT1, FBXO7, mePHGDH (R236me1), and Ki-67. The tissue microarray (TMA) comprising 45 HCC tissues and paired adjacent normal tissues (obtained from West China Hospital, Chengdu, with informed written consent from patients; clinical characteristics of HCC patients were shown in Supplementary Table 1) were subjected to IHC staining to evaluate the protein levels of PRMT1, FBXO7, and mePHGDH (R236me1). The IHC staining was performed following the previous report[15]. Briefly, the paraffin-embedded slides (5 μm) were deparaffinized, rehydrated, and blocked with 3% H$_2$O$_2$. Antigen retrieval was then conducted using a citrate buffer. The sections were then blocked with 10% goat serum at 37 °C for 1 h, and subsequently incubated with the indicated primary antibodies at 4 °C overnight. The slides were treated with MaxVision HRP-Polymer anti-Mouse/Rabbit IHC Kit (MXB Biotechnology, 5010), stained with diaminobenzidine (MXB Biotechnology, 0031), and finally counterstained with Mayer's hematoxylin (MXB Biotechnology, CTS1096). The intensity of IHC staining was graded as negative (0), weak (1), moderate (2), and strong (3). H-score = (1 × % of weak staining) + (2 × % moderate staining) + (3 × % strong staining). The staining intensity and H-score were evaluated independently by two investigators.

## Statistics and reproducibility

Statistical analyses were performed using GraphPad Prism 8.0 software. Two-tailed Student's $t$-test was performed to compare the variables between the two groups. Two-way ANOVA with Bonferroni correction was used to analyze statistical differences for tumor/cell growth rate or protein half-life. Pearson correlation and linear regression were used to determine the correlation in clinical samples. Kaplan–Meier survival curve and the log-rank test were conducted to analyze the survival. All experiments were repeated independently three times with similar results. All data represent the mean ± SD of three independent experiments, unless otherwise specified.

## Reporting summary

Further information on research design is available in the Nature Portfolio Reporting Summary linked to this article.

# Data availability

The proteomics data generated in this study have been deposited in the ProteomeXchange Consortium (http://proteomecentral.proteomexchange.org) via the iProX partner repository with the dataset identifier PXD045015 and PXD049069. Source data are provided with this paper. The remaining data are available within the

Article, Supplementary Information or Source Data file. Source data are provided with this paper.

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

## Acknowledgements

This work was supported by the Chinese NSFC No. 82073081 (K.W.), No. 82373122 (K.W.), No. 82002963 (L.L.), No. 82373336 (J.J.), No. 82003098 (J.J.) and No. 82303238 (W.Y.); Guangdong Basic and Applied Basic Research Foundation No. 2019B030302012 (K.W.).

## Author contributions

L.L., X.W., J.F., L.D., M.W., Y.Z., S.L., and W.Y. conducted the experiments and analyzed the data. J.J. and K.W. conceived and supervised the study. L.L., J.J., and K.W. wrote the manuscript with comments from all authors.

## Competing interests

The authors declare no competing interests.
