## [Peer Review File · Nature Communications]

FBXO7 Ubiquitinates PRMT1 to Suppress Serine Synthesis and Tumor Growth in Hepatocellular CarcinomaEditorial Note: Parts of this Peer Review File have been redacted as indicated to maintain the confidentiality of unpublished data.

REVIEWER COMMENTS

Reviewer #1 (Remarks to the Author):

Thank you for asking me to review the manuscript 'FBXO7 Ubiquitinates PRMT1 to Suppress Serine Synthesis and Tumor Growth in Hepatocellular Carcinoma' by Luo et al. The authors have sought to build on their recent Nature Comms paper by Wang et al demonstrating the importance of PHGDH-mediated serine metabolism on the pathogenesis of hepatocellular carcinoma.

In this paper they have sought to define the regulation of PHGDH, identifying PRMT1 and its post-translational regulation by FBXO7, through proteasomal-mediated degradation as a key player in this metabolic pathway. They have conducted a convincing set of experiments demonstrating the binding and ubiquitination of PRMT1. They have demonstrated this in several cells lines and then demonstrated the effect on serine metabolism and then tumour growth through xenograft studies in mice. Subsequent human relevance is limited to immunostaining of a tissue array and mining of a publicly-available mRNA-Sequencing dataset from HCC samples.

The experiments are well conducted with appropriate controls. The data are presented clearly and the written English is good.

I have the following comments:

Major points

1. The demonstration of enhanced ubiquitination of GFP-PRMT1 in figure 2D is not convincing. Could you authors repeat or potentially use a different cell line to demonstrate? Similarly, it is not clear whether the Delta-F-box mutant of FBXO7 (Figure 2D) corresponds to one of the mutants in figure 1J; please elaborate in the text or legend. It is not clear which shFBXO7 that they have used in several experiments (Figure 2C / 3E). This is important as in Figure 3C they are using Huh7 cells transduced with shFBXO7, but no demonstration of the effectiveness of the knockdown in these blots.
2. Figure 3F is uninterpretable as it is too small with no guide in the text or legend as to which peak / fragment we should be looking at. Have they conducted quantitative MS in any of their conditions to determine the relative change in ubiquitination, as a second modality other than immunoblotting?
3. The experiment that is missing from figure 4 is the demonstration that over-expression of the K37R mutant of PRMT1 prevents the reduction in R236 methylation and enzymatic function of PHGDH associated with FBXO7 over-expression. Can they demonstrate this by blotting or functional analysis??
4. I think the diagram in figure 7M is confusing. It suggests that FBXO7 and PHGDH are interchangeable in the binding to PRMT1; this should be changed to make it clearer that FBXO7 is inhibiting the action of PRMT1 on PHGDH as they have not demonstrated competitive binding. Secondly, they have spelt it 'FBOX7'.
5. Could they conduct further analysis of publicly-available datasets to determine which transcriptional subtype of HCC has elevated expression of FBXO7 or PRMT1? One could

hypothesise that this would be limited to one sub-group of patients and this data would suggest potential stratification of subsequent trials targeting this pathway.

Minor points

1. Demonstration of dose-dependency of proteasome inhibitors only checked with one drug. I would suggest that the authors confirm with a second agent such as lactacystin or bortezomib.
2. Fig 1H / J – add protein titles so it makes it clearer to the reader which protein you are making truncation mutants of.
3. Figure 6. It is not clear on the figure that the left and right panels reflect normal and serine / glycine depleted media. Please add to the figure to make clearer.
4. It is interesting that there is not a single described patient with a cancer-associated SNV of K37 in PRMT1 in the COSMIC database. One would hypothesise that this would generate an advantageous phenotype and therefore be selected for. Perhaps the authors could add to the discussion about why this might be?

Reviewer #2 (Remarks to the Author):

This study focuses on serine synthesis regulation in hepatocellular carcinoma (HCC). The authors previously showed that PRMT1, upregulated in HCC, activates PHGDH, a key enzyme in serine synthesis. Here they investigate the regulation of PRMT1. The E3 ubiquitin ligase FBXO7 promotes PRMT1 degradation, inhibiting serine synthesis. FBXO7 interacts with PRMT1, leading to its ubiquitination and degradation, disrupting serine synthesis. This downregulation of PRMT1 results in reduced serine synthesis, elevated reactive oxygen species, and impaired HCC cell growth. Clinical data support the clinical relevance indicating that FBXO7 is downregulated in HCC tissues, inversely correlated with PRMT1 expression and PHGDH methylation.

The study unveils a novel mechanism involving FBXO7, an E3 ubiquitin ligase, which modulates PRMT1 ubiquitylation. The data provides solid biochemical evidence of a novel regulation by FBXO7 of PRMT1 stability. Nonetheless, a key weakness lies in the limited biological significance of the observed effects, both in in vivo tumor growth and in vitro cell proliferation, which appear marginal and inconclusive. This raises questions regarding the broader biological implications of the described biochemical axis.

In addition to this, several specific points should be addressed:

- The biological consequences of FBXO7 are not clear. A thorough analysis of the biological implications of the observed GSH and ROS imbalances within the biological system would be essential. Questions that need addressing include whether FBXO7-overexpressing cells exhibit cell death and cell cycle arrest due to ROS accumulation, whether overexpression of a PRMT1 K37R mutant can rescue these effects, and whether supplementation of antioxidants in vitro can restore the observed effects.

- In addition to the point above, addressing the biological significance in vivo, there will still be uncertainty in the in vivo effects, who appeared generally marginal.

The study lacks a clear elucidation of the causative relationships within the FBXO7/PRMT1 axis. Assessing how FBXO7 overexpression impacts tumor growth and whether PRMT1 K37R can rescue the effects of FBXO7 overexpression in vivo is critical.

- A notable gap in the study is the absence of a comprehensive metabolic analysis to directly assess the effects of FBXO7. Understanding whether the alterations in serine/glycine biosynthesis result from direct or indirect consequences on alternative metabolic pathways is crucial. Conducting global metabolomics and metabolic tracing experiments, especially with marked glucose in serine/glycine-deprived and non-deprived media, could provide valuable insights into global metabolic changes and possibly open to broader biological significance.

- Investigating the effects of PRMT1 K37R on PHGDH methylation and activity and whether it preserves or restores the effects of FBXO7 overexpression would also provide stronger evidence in support of the described axis.

In conclusion, while the study introduces a novel regulatory aspect of FBXO7 on PRMT1 ubiquitylation, its limited biological significance and gaps in the mechanistic understanding warrant further investigation.

Reviewer #3 (Remarks to the Author):

In this paper on serine synthesis regulation in cancer, Luo et al. describe how F-box-only protein 7 (FBXO7), an E3 ubiquitin ligase, inhibits serine synthesis by promoting the ubiquitin-proteasome degradation of protein arginine methyltransferase 1 (PRMT1) in hepatocellular carcinoma (HCC). Previously, they had already shown that PRMT1 is upregulated in HCC. PRMT1 methylates phosphoglycerate dehydrogenase (PHGDH) at arginine 236, thereby activating PHGDH and promoting serine synthesis. Nevertheless, the mechanisms underlying overexpression of PRMT1 and regulation of the PRMT1-PHGDH axis in HCC remains unknown.

Here, the authors unravel the details of this mechanism. In a series of elegant experiments including all relevant controls, the authors convincingly show that FBXO7 directly interacts with PRMT1, leading to the lysine 37 ubiquitination and subsequent degradation of PRMT1. FBXO7-mediated PRMT1 downregulation then decreases the extent to which PHGDH is methylated on its target arginine residue, resulting in impaired serine synthesis, accumulation of reactive oxygen species (ROS) levels, and inhibition of HCC cell growth.

The authors have gathered a substantial amount of evidence to support these claims, including in vitro biochemical experiments and clinical data, further strengthening their hypothesis.

Overall, the design of the experiments and the levels of detail are very solid. The data convincingly show that FBXO7 suppresses PHGDH R236 methylation and catalytic activity by downregulating PRMT1 in HCC cells.

I have no major comments, only a few minor comments listed below in bullet points:

- P5: The authors state that “[...] we and others have revealed that PRMT1 is frequently

overexpressed in HCC to promote tumor growth [...]”. Is it really overexpressed or is it just degraded to a lesser extent upon expression? As the mRNA levels seem not to change, I assume it is the latter.

- P5: Although the authors have used a smart in silico screening to identify interaction partners that ultimately provided them the link to FBXO7, I'm wondering why the authors have not performed an unbiased interaction screen based on affinity purification coupled to mass spectrometric identification of interaction partners themselves. The advantage would be that PRMT1 interactions in the relevant cell system (Huh7 and PLC/PRF/5) could be studied this way. I'm not sure whether BioGRID contains interactome data for this specific cell line.

- P7: “[...] as shown in Figure 2A, the upregulated protein level of PRMT1 caused by FBXO7 KD was not observed in MG132-treated HCC cells [...]”. The Western blots for the PLC/PRF/5 cells are convincing in this respect, but the Huh7 less so. It's hard to see the lack of upregulated levels in the MG132 treated FBXO7 KD cells. Have the authors considered more quantitative approaches to determine PRMT1 and FBXO7 levels, such as quantitative targeted mass spectrometry (PRM)?

- P8: In Fig 3C, although the high MW smear as observed in the WT lane is certainly not observed in the K37R cells, the intensity of the bands / smear in the HA IB seems to be increased in the K37R mutant with respect to the WT (minus FLAG-FBXO7). Do the authors have an explanation for this? Also, even though the ubiquitination of the protein results in a smear on the blot, distinct bands are observed. Is it possible, based on these results, to hypothesize on the substrate being mono- or poly-ubiquitinated?

- In the fragmentation mass spectrum in Fig 3F, no (or, at most, noise intensity level) fragment ion peaks are observed around the Lysine residue with the diGly remnant. However, given the fact that the K-D bond is not cleaved and the total peptide mass corresponds to that of the modified peptides, the data convincingly shows that K37 is ubiquitinated. Was the PHGDH R236 methylation site also confirmed by MS, like the K37 ubiquitination site of PRMT1?

- In Fig 4G, the statistical (in)significance indicators for the shPRMT1 shScramble vs shFBXO7 are missing (idem for Figs 5A and B).

- Have the authors considered including experiments with a mutant PHGDH that lacks the R236 target methylation site? If so, why were these not included in the manuscript?

- The model in Fig 7M suggests that low levels of FBXO7 lead to ROS accumulation (I suppose that the spiked balloon with the text 'ROS' means 'accumulation of ROS'). However, in Fig 5H, the authors show that the depletion of FBXO7 causes a reduction of ROS levels and that the depletion of PRMT1 leads to increased ROS levels. Also, ectopic FBXO7 expression promotes ROS accumulation in control cells and has no obvious effect in PRMT1 KD cells (Fig 5I). This suggests that FBXO7 induces oxidative stress by downregulating PRMT1 in HCC cells. Therefore, the description of the results and the cartoon in Fig 7M seem to contradict each other. Could the authors comment on that?

Reviewer #4 (Remarks to the Author):

This manuscript entitled "FBXO7 Ubiquitinates PRMT1 to Suppress Serine Synthesis and Tumor Growth in Hepatocellular Carcinoma" by Li Luo et al identified a tumor-suppressive role of FBXO7 in HCC. The authors first showed that FBXO7 is a PRMT1-interacting partner by in vitro cell line-based assay. Next they demonstrated that FBXO7 ubiquitinates PRMT1 and identified the K37 in PRMT1 is the key ubiquitination site. Further functional studies showed that FBXO7 suppressed PHGDH activity by downregulating PRMT1-mediated PHGDH methylation (R236me1), causing the inhibition of serine synthesis and HCC repression. Overall, the manuscript is well-designed and organized, the evidence supports the conclusion, and the findings will be of broad interest to the Liver cancer field.

Major concern:

1. in Figure 1I, by IP assay, it seems the PRMT1 protein catalytic fragment (23-162) is the key for the interaction with FBXO7, however, this is not precisely demonstrated in the manuscript. In the Fig. 1K, which part of FBXO7 is required for its interaction with PRMT1? this is not well defined in its current form. Moreover, it would be good to show the co-localization for their interaction, e.g. by immunofluorescent staining as a surrogate.
2. Although the authors provided both the negative correlation and xenograft studies using clinical biopsies samples, further experimental demonstration of reciprocal interaction between FBXO7 and PRMT1, and perhaps the presence of PRMT1 ubiquitination site, in primary HCC tumors will strengthen the conclusion.

Minor concerns:

P6, "Figure 2H and 2I)" should allude to Fig. 1H, 1I.

P7, Figure 2J and 2K should mention "Figure 1J and 1K"

P34, in Figure 3C: top panel: I can see GFPtagged PRMT1 was ubiquitinated as visualized by the smear bands in second WT and K82R/202R/325R samples. However, in the bottom IB:GFP loading control, the GFP bands showed as distinctive single bands (70 kD), should not they appear as smear bands?

P34, in Figure 3E: Please elaborate how the band intensity was quantified, because it seems to me the data differences at 6/12/24 hr were not that different as presented.

Point-by-point response

Reviewer's Comments:

Reviewer #1 (Remarks to the Author)

Thank you for asking me to review the manuscript 'FBXO7 Ubiquitinates PRMT1 to Suppress Serine Synthesis and Tumor Growth in Hepatocellular Carcinoma' by Luo et al. The authors have sought to build on their recent Nature Comms paper by Wang et al demonstrating the importance of PHGDH-mediated serine metabolism on the pathogenesis of hepatocellular carcinoma.

In this paper they have sought to define the regulation of PHGDH, identifying PRMT1 and its post-translational regulation by FBXO7, through proteasomal-mediated degradation as a key player in this metabolic pathway. They have conducted a convincing set of experiments demonstrating the binding and ubiquitination of PRMT1. They have demonstrated this in several cells lines and then demonstrated the effect on serine metabolism and then tumour growth through xenograft studies in mice. Subsequent human relevance is limited to immunostaining of a tissue array and mining of a publicly-available mRNA-Sequencing dataset from HCC samples.

The experiments are well conducted with appropriate controls. The data are presented clearly and the written English is good.

I have the following comments:

Major points

1. The demonstration of enhanced ubiquitination of GFP-PRMT1 in figure 2D is not convincing. Could you authors repeat or potentially use a different cell line to demonstrate? Similarly, it is not clear whether the Delta-F-box mutant of FBXO7

(Figure 2D) corresponds to one of the mutants in figure 1J; please elaborate in the text or legend. It is not clear which shFBXO7 that they have used in several experiments (Figure 2C / 3E). This is important as in Figure 3C they are using Huh7 cells transduced with shFBXO7, but no demonstration of the effectiveness of the knockdown in these blots.

Response: Thank you for your valuable comments. Following your suggestion, we used Huh7 cells instead of HEK293T cells to conduct the ubiquitination assay. As shown in Fig. 2e, Expression of FLAG-FBXO7 WT, but not its enzymatically dead Δ F-box mutant, prominently increased the ubiquitination level of GFP-PRMT1 in Huh7 cells (Page 7, Lines 9-13 in the revised manuscript with track changes).

The Δ F-box mutant of FBXO7 in Fig. 2d (Fig 2e in the revised version) means that the F-box domain (aa 332-375) was deleted, which was commonly used as an enzymatically dead mutant of the E3 ligase FBOX7 (Stott SR *et al.*, *J Pathol.* 2019;249(2):241-54). We have elaborated this in the figure (Fig. 2d in the revised version) and figure legend.

Figure 2. d Schematic representation of full-length (FL) FBXO7 and enzymatically dead mutant (deletion of F-box domain, shown as Δ F-box). **e** GFP-PRMT1 and HA-ubiquitin (HA-Ub) were co-expressed with FLAG-FBXO7 WT or enzymatically dead Δ F-box mutant in Huh7

cells. After MG132 (25 μ M, 6 h) treatment, IP was performed with GFP antibody, followed by immunoblotting with indicated antibodies.

In our study, sh*FBXO7* #2 (targeting 3'UTR) was used in Fig. 2c, 3c, and 3e, as well as Fig. 4, Fig. 5, and Fig. 6. The knockdown effectiveness of sh*FBXO7* #2 was shown in Fig. 2a and 2c. We also repeated Fig. 3c and detected FBXO7 level in whole cell lysates to demonstrate the efficiency of *FBXO7* KD as well as *FBXO7* rescue in Huh7 cells. Similarly, the sh*PRMT1* was also corrected as sh*PRMT1* #2 (targeting 3'UTR) in the related figures.

Figure 3. c GFP-PRMT1 (WT or indicated KR mutants) was co-expressed with HA-Ub in *FBXO7* KD Huh7 cells rescued with or without FLAG-FBXO7. After MG132 (25 μ M, 6 h) treatment, IP was performed with GFP antibody, followed by immunoblotting with indicated antibodies.

2. Figure 3F is uninterpretable as it is too small with no guide in the text or legend as to which peak / fragment we should be looking at. Have they conducted quantitative MS in any of their conditions to determine the relative change in ubiquitination, as a second modality other than immunoblotting?

Response: We apologize for the small size of Fig. 3f with no guide. This image has been optimized to make the b-ions and y-ions clearer (Fig. 3f). The ubiquitinated lysine 37

(K37, y_5^{1+} ion with an additional shift of 114.1 Da) was highlighted.

Figure 3. f Mass spectrometry identification of K37 ubiquitination of PRMT1 (workflow 1 in Figure R1).

We also performed parallel reaction monitoring (PRM)-based quantitative proteomics approach to quantitate PRMT1 K37 ubiquitination level following FBXO7 overexpression. Unfortunately, we failed to identify K37 ubiquitination of PRMT1 protein during PRM analysis probably due to the relatively low abundance of PRMT1 protein for PRM proteomics analysis. Because of the quantitative purpose, antibody-based affinity purification and enrichment of PRMT1 protein is not allowed in PRM analysis (Fig. R1, workflow 3 for PRM analysis). Compared with the workflows for ubiquitination site identification with antibody enrichment of PRMT1 protein (Fig. R1, workflow 1 and workflow 2), relatively low PRMT1 protein level was subjected to PRM proteomics analysis, which might lead to the failure for the recognition of PRMT1 K37 ubiquitination by mass spectrometry.

[Redacted]

Figure R1. Schematics of the workflows for identifying the ubiquitination sites of PRMT1 by mass spectrometry analysis (workflow 1 and workflow 2), and for quantitating PRMT1 K37

ubiquitination level in cells overexpressing FBXO7 compared with empty vector by PRM-based quantitative proteomics analysis (workflow 3).

Despite the failure of PRMT1 K37 quantitation by PRM-based quantitative proteomics, we indeed identified K37 ubiquitination of PRMT1 using two different sample preparation workflows for MS analysis (workflow 1, Fig. 3f; and workflow 2, Fig. R2). In addition, our *in vivo* ubiquitination assay combined with lysine mutation convincingly demonstrated that FBXO7 mediates PRMT1 K37 ubiquitination and promotes ubiquitin-mediated degradation of PRMT1 in HCC cells (Fig. 2 and 3).

Figure R2. f Mass spectrometry identification of K37 ubiquitination of PRMT1 (workflow 2 workflow 1 in Figure R2).

3. The experiment that is missing from figure 4 is the demonstration that over-expression of the K37R mutant of PRMT1 prevents the reduction in R236 methylation and enzymatic function of PHGDH associated with FBXO7 over-expression. Can they demonstrate this by blotting or functional analysis??

Response: Following your valuable suggestions, we examined PHGDH methylation level and PHGDH activity in HCC cells overexpressing FLAG-FBXO7 and GFP-PRMT1-WT or K37R. As shown in Fig. 4d, enforced expression of FBXO7 led to

obviously decreased PHGDH methylation level. Overexpressing K37R mutant, but not wild-type PRMT1, obviously rescued the decreased level of PHGDH methylation in FBXO7-overexpressing cells. In addition, FBXO7 overexpression resulted in reduced PHGDH activity in HCC cells, which was markedly restored by K37R mutant of PRMT1, but not its wild-type (Fig. 4h). These results demonstrate that FBXO7 suppresses PHGDH methylation and activity by downregulating PRMT1 in HCC cells (Page 8, Lines 26-28; and Page 9, Lines 6-8 in the revised manuscript).

Figure 4. **d** PHGDH was immunoprecipitated in cells overexpressing FLAG-FBXO7 and GFP-PRMT1-WT or K37R, followed by immunoblotting with indicated antibodies. **h** Endogenous PHGDH was immunoprecipitated in cells overexpressing FLAG-FBXO7 and GFP-PRMT1-WT or K37R, followed by measurement of PHGDH activity.

4. I think the diagram in figure 7M is confusing. It suggests that FBXO7 and PHGDH are interchangeable in the binding to PRMT1; this should be changed to make it clearer that FBXO7 is inhibiting the action of PRMT1 on PHGDH as they have not demonstrated competitive binding. Secondly, they have spelt it 'FBOX7'.

Response: We apologize for this confusing diagram. According to your kind suggestion, we have simplified this schematic model to make it clearer that FBXO7 promotes the ubiquitin-mediated degradation of PRMT1, leading to decreased PHGDH methylation. In addition, we have corrected 'FBOX7' as 'FBXO7' (Supplementary Fig. 8 in the revised version).

Supplementary Fig. 8. Schematic model for the mechanism of FBXO7-PRMT1-PHGDH axis in regulating serine synthesis, oxidative stress, and HCC growth. The E3 ubiquitin ligase FBXO7 promotes PRMT1 degradation by lysine 37 ubiquitination, leading to reduction of PHGDH arginine methylation and catalytic activity, suppression of serine synthesis, accumulation of ROS, and inhibition of HCC growth.

5. Could they conduct further analysis of publicly-available datasets to determine which transcriptional subtype of HCC has elevated expression of FBXO7 or PRMT1? One could hypothesize that this would be limited to one sub-group of patients and this data would suggest potential stratification of subsequent trials targeting this pathway.

Response: Thanks for your valuable suggestions. Given that the tumor suppressor gene *TP53* was mutated in 31% of HCC patients, and p53 target gene expression signature (p53 signature) correlating with patient survival was recently developed (Cancer Genome Atlas Research Network. *Cell*. 2017;169(7):1327-41). We thus stratified 186 HCC patients in TCGA database by p53 signature, which showed that patients with low p53 signature exhibited reduced overall survival (Supplementary Fig. 7c). Interestingly, the p53 signature was positively correlated with the mRNA level of *FBXO7* (Supplementary Fig. 7d), suggesting that *FBXO7* expression may be linked with the sub-group of patients with different p53 signature. Therefore, therapeutic strategies targeting FBXO7-PRMT1-PHGDH axis might be applicable for HCC patients with low p53 signature, which requires further investigation (Page 12, Lines 27-29; Page 13, Lines 1-6 in the revised manuscript with track changes).

Supplementary Figure 7. c Overall survival of HCC patients based on p53 signature ($n = 186$ samples) in TCGA database. **d** Pearson correlation test analyzing the relationship between FBXO7 level and p53 signature in HCC tissues ($n = 186$ samples).

Minor points

1. Demonstration of dose-dependency of proteasome inhibitors only checked with one drug. I would suggest that the authors confirm with a second agent such as lactacystin or bortezomib.

Response: Thanks for your kind suggestion. We used bortezomib to confirm the effect of proteasomal inhibitors on PRMT1 protein level. In line with the observation in MG132-treated cells, a dose-dependent increase of PRMT1 protein level was also observed following bortezomib treatment (Supplementary Fig. 1c) (Page 5, Lines 4-7 in the revised manuscript with track changes). In addition, immunoblotting analysis using two different commercial PRMT1 antibodies demonstrated that the upregulated protein level of PRMT1 caused by *FBXO7* KD was not observed in bortezomib-treated HCC cells (Supplementary Fig. 3h) (Page 6, Lines 23-28; and Page 7, Line 1 in the revised manuscript with track changes). These data suggest that FBXO7-mediated downregulation of PRMT1 protein level is attributed to ubiquitin-proteasome degradation.

Supplementary Figure 1. c Immunoblotting analysis of PRMT1 (anti-PRMT1 antibody: Abcam, ab190892) in Huh7 and PLC/PRF/5 cells treated with bortezomib at indicated concentrations for 5 h.

Supplementary Figure 3. h Immunoblotting analysis of PRMT1 (anti-PRMT1 antibody: Abcam, ab190892; Proteintech, 11279-1-AP) and FBXO7 in *FBXO7* KD cells treated with bortezomib (500 nM, 5 h).

2. Fig 1H / J – add protein titles so it makes it clearer to the reader which protein you are making truncation mutants of.

Response: Following your valuable suggestion, we have added protein titles in Fig. 1h and 1j (Fig 1h and 1k in the revised version).

3. Figure 6. It is not clear on the figure that the left and right panels reflect normal and serine / glycine depleted media. Please add to the figure to make clearer.

Response: Thanks for your kind suggestion. We have added the CM medium (complete medium) or -SG medium (serine/glycine depleted medium) in each panel in Fig. 6.

4. It is interesting that there is not a single described patient with a cancer-associated SNV of K37 in PRMT1 in the COSMIC database. One would hypothesize that this would generate an advantageous phenotype and therefore be selected for. Perhaps the authors could add to the discussion about why this might be?

Response: Thanks for your critical comments. Our study demonstrates that *FBXO7* plays a tumor suppressive role in HCC, and the mRNA level of *FBXO7* is positively correlated with p53 signature (Supplementary Fig. 7d) (Page 12, Lines 27-29; Page 13, Lines 1-6 in the revised manuscript with track changes). It is possible that in most HCC patients, the mRNA level of *FBXO7* is downregulated due to the mutation of *TP53* or other frequently mutated tumor suppressor genes. In this case, PRMT1 will be sufficiently upregulated due to less *FBXO7*-mediated degradation, and there might be no need to select K37 mutant for HCC development. We have added these descriptions in the discussion section (Page 14, Lines 21-25 in the revised manuscript with track changes).

Supplementary Figure 7. c Overall survival of HCC patients based on p53 signature ($n = 186$ samples) in TCGA database. **d** Pearson correlation test analyzing the relationship between *FBXO7* level and p53 signature in HCC tissues ($n = 186$ samples).

Reviewer #2 (Remarks to the Author)

This study focuses on serine synthesis regulation in hepatocellular carcinoma (HCC). The authors previously showed that PRMT1, upregulated in HCC, activates PHGDH,

a key enzyme in serine synthesis. Here they investigate the regulation of PRMT1. The E3 ubiquitin ligase FBXO7 promotes PRMT1 degradation, inhibiting serine synthesis. FBXO7 interacts with PRMT1, leading to its ubiquitination and degradation, disrupting serine synthesis. This downregulation of PRMT1 results in reduced serine synthesis, elevated reactive oxygen species, and impaired HCC cell growth. Clinical data support the clinical relevance indicating that FBXO7 is downregulated in HCC tissues, inversely correlated with PRMT1 expression and PHGDH methylation.

The study unveils a novel mechanism involving FBXO7, an E3 ubiquitin ligase, which modulates PRMT1 ubiquitylation. The data provides solid biochemical evidence of a novel regulation by FBXO7 of PRMT1 stability. Nonetheless, a key weakness lies in the limited biological significance of the observed effects, both in *in vivo* tumor growth and *in vitro* cell proliferation, which appear marginal and inconclusive. This raises questions regarding the broader biological implications of the described biochemical axis.

Response: Thank you for your critical comments. To strengthen our conclusion and broaden the biological significance regarding FBXO7-mediated PRMT1 K37 ubiquitination, we stably expressed FLAG-FBXO7 and GFP-PRMT1-WT/K37R in Huh7 and PLC/PRF/5 cells. We found that FLAG-FBXO7 overexpression reduced PHGDH methylation and activity, suppressed serine synthesis, and induced ROS accumulation, leading to cell cycle arrest, apoptosis induction, as well as growth inhibition of HCC cells *in vitro*. In addition, the increase of oxidative stress, induction of apoptosis, and inhibition of growth of HCC cells caused by FBXO7 overexpression were further confirmed in HCC tumor xenografts *in vivo*. All of these observations in FBXO7-overexpressing cells were markedly restored by PRMT1 K37R mutant, but not its wild-type. Moreover, ROS accumulation, apoptosis induction, and growth inhibition caused by FBXO7 overexpression could be reversed by treatment with the antioxidant N-acetyl cysteine (NAC). These related results and figures can be found in detail in the following responses. We believe that the revised version of this study will provide broader biological implications of the described biochemical axis.

In addition to this, several specific points should be addressed:

- The biological consequences of FBXO7 are not clear. A thorough analysis of the biological implications of the observed GSH and ROS imbalances within the biological system would be essential. Questions that need addressing include whether FBXO7-overexpressing cells exhibit cell death and cell cycle arrest due to ROS accumulation, whether overexpression of a PRMT1 K37R mutant can rescue these effects, and whether supplementation of antioxidants in vitro can restore the observed effects.

Response: Thank you for your critical comments. We stably expressed FLAG-FBXO7 and GFP-PRMT1-WT/K37R in Huh7 and PLC/PRF/5 cells to examine the biological consequences of FBXO7-mediated PRMT1 ubiquitination and degradation. As shown in Fig. 5f, FBXO7 overexpression promoted ROS accumulation in Huh7 and PLC/PRF/5 cells, which was significantly compromised by introducing K37R mutant, but not wild-type, of PRMT1. In addition, ROS accumulation in FBXO7-overexpressing cells with serine/glycine starvation was markedly attenuated by NAC (Fig. 5g). These data demonstrate that FBXO7 promotes ROS accumulation and oxidative stress by downregulating PRMT1 in HCC cells (Page 10, Lines 16-20 in the revised manuscript with track changes).

Figure 5. **f** ROS levels in cells overexpressing FLAG-FBXO7 and GFP-PRMT1-WT or K37R cultured in CM or -SG medium. **g** ROS levels in FLAG-FBXO7-overexpressing cells cultured in -SG medium, in the presence or absence of NAC (5 mM).

We further determined the role of FBXO7-mediated PRMT1 ubiquitination and

degradation in HCC growth. The suppressed growth of HCC cells caused by FBXO7 overexpression was significantly rescued by K37R mutant of PRMT1, but not wild-type PRMT1 (Fig. 6b). Moreover, FBXO7 overexpression led to cell cycle arrest (Supplementary Fig. 6b) and apoptosis induction (as evidenced by increased cleavage of caspase 3; Supplementary Fig. 6c) in HCC cells grown in serine/glycine-free (-SG) medium, which was prominently restored by K37R mutant of PRMT1, but not wild-type PRMT1. Notably, the decreased cell growth and increased caspase 3 cleavage in FBXO7-overexpressing cells with serine/glycine starvation were markedly recovered by NAC treatment (Fig. 6c, d). These data indicate that FBXO7 induces apoptosis and inhibits the growth of HCC cells by downregulating PRMT1 and promoting ROS accumulation (Page 11, Lines 5-13 in the revised manuscript with track changes).

Figure 6. b Growth rates of cells overexpressing FLAG-FBXO7 and GFP-PRMT1-WT or K37R grown in CM or -SG medium. **c** Growth rates of FLAG-FBXO7-overexpressing cells grown in -SG medium in the presence or absence of NAC. **d** Immunoblotting analysis of cleaved caspase 3 and caspase 3 in FLAG-FBXO7-overexpressing cells grown in -SG medium in the presence or absence of NAC.

Supplementary Figure 6. b Cell cycle distribution of Huh7 and PLC/PRF/5 cells overexpressing FLAG-FBXO7 and GFP-PRMT1-WT or K37R cultured in CM or -SG medium. **c** Immunoblotting analysis of cleaved caspase 3 and caspase 3 in cells overexpressing FLAG-FBXO7 and GFP-PRMT1-WT or K37R cultured in CM or -SG medium.

- In addition to the point above, addressing the biological significance *in vivo*, there will still be uncertainty in the *in vivo* effects, who appeared generally marginal. The study lacks a clear elucidation of the causative relationships within the FBXO7/PRMT1 axis. Assessing how FBXO7 overexpression impacts tumor growth and whether PRMT1 K37R can rescue the effects of FBXO7 overexpression *in vivo* is critical.

Response: Thank you for your critical comments. Following your valuable suggestions, we generated a mouse xenograft model by subcutaneously inoculating Huh7 cells overexpressing FLAG-FBXO7 and GFP-PRMT1-WT or K37R into nude mice, and then examined the effect of FBXO7-mediated PRMT1 ubiquitination in tumor growth. As shown in Fig. 6g, h, overexpression of FBXO7 inhibited the growth of xenografts from mice fed with a +SG diet. A further decrease in tumor size and weight was observed when mice were fed with a -SG diet (Fig. 6g, h). The inhibition effect of FLAG-FBXO7 on tumor growth was markedly recovered by K37R mutant of PRMT1, but not wild-type PRMT1 (Fig. 6g, h). Notably, tumor xenograft with FBXO7 overexpression showed decreased intensity of Ki-67 staining, accompanied by increased intensity of 8-oxo-dG (a marker of oxidative stress) and cleaved caspase 3 (an apoptosis marker) staining, all of which were obviously rescued by expressing PRMT1 K37R mutant, but not PRMT1 WT (Fig. 6i, Supplementary Fig. 6g). Taken

together, these results demonstrate that FBXO7-mediated PRMT1 ubiquitination promotes oxidative stress, induces apoptosis, and inhibits HCC xenografts growth *in vivo* (Page 11, Lines 27-29; Page 12, Lines 1-9 in the revised manuscript with track changes).

Figure 6. g, h Huh7 cells overexpressing FLAG-FBXO7 and GFP-PRMT1-WT or K37R were subcutaneously inoculated into nude mice fed with a +SG or -SG diet. Tumor image (**g**) and weight (**h**) were shown. **i**, Representative images of IHC staining for Ki-67, 8-oxo-dG, and cleaved caspase 3 in tumor xenografts in **g**. Scale bars, 50 μ m.

Supplementary Figure 6. **g** Quantitative analysis of IHC staining for Ki-67 of tumor xenografts from nude mice inoculated with Huh7 cells overexpressing FLAG-FBXO7 and GFP-PRMT1-WT or K37R and fed with a +SG or -SG diet.

- A notable gap in the study is the absence of a comprehensive metabolic analysis to directly assess the effects of FBXO7. Understanding whether the alterations in serine/glycine biosynthesis result from direct or indirect consequences on alternative metabolic pathways is crucial. Conducting global metabolomics and metabolic tracing experiments, especially with marked glucose in serine/glycine-deprived and non-deprived media, could provide valuable insights into global metabolic changes and possibly open to broader biological significance.

Response: Thank you very much for your critical comment. Following your valuable suggestion, we used U-¹³C]-glucose to trace the *de novo* synthesis of serine and glycine (Fig. 5a). As shown in Fig. 5b, c, the incorporation of ¹³C from glucose into serine and glycine was markedly diminished by enforced expression of FLAG-FBXO7, which could be rescued by overexpressing PRMT1 K37R mutant, but not PRMT1 WT. These data indicate that FBXO7 inhibits serine synthesis by downregulating PRMT1 (Page 9, Lines 21-26 in the revised manuscript with track changes).

Figure 5. a Schematic of U-¹³C]-glucose incorporation into serine and glycine in cells. **b, c** Incorporation of U-¹³C]-glucose carbon into serine (**b**) and glycine (**c**) detected by LC-MS/MS in cells overexpressing FLAG-FBXO7 and GFP-PRMT1-WT or K37R.

- Investigating the effects of PRMT1 K37R on PHGDH methylation and activity and

whether it preserves or restores the effects of FBXO7 overexpression would also provide stronger evidence in support of the described axis.

Response: Thank you for your kind suggestions. We examined PHGDH methylation level and PHGDH activity in HCC cells overexpressing FLAG-FBXO7 and GFP-PRMT1-WT or K37R. As shown in Fig. 4d, enforced expression of FBXO7 led to obviously decreased PHGDH methylation level. Overexpressing K37R mutant, but not wild-type PRMT1, rescued the decreased level of PHGDH methylation in FBXO7-overexpressing cells. In addition, FBXO7 overexpression resulted in reduced PHGDH activity in HCC cells, which was markedly restored by K37R mutant of PRMT1, but not its wild-type (Fig. 4h). These results demonstrate that FBXO7 suppresses PHGDH methylation and activity by downregulating PRMT1 in HCC cells (Page 8, Lines 26-28; and Page 9, Lines 6-8 in the revised manuscript).

Figure 4. d PHGDH was immunoprecipitated in cells overexpressing FLAG-FBXO7 and GFP-PRMT1-WT or K37R, followed by immunoblotting with indicated antibodies. **h** Endogenous PHGDH was immunoprecipitated in cells overexpressing FLAG-FBXO7 and GFP-PRMT1-WT or K37R, followed by measurement of PHGDH activity.

In conclusion, while the study introduces a novel regulatory aspect of FBXO7 on PRMT1 ubiquitylation, its limited biological significance and gaps in the mechanistic understanding warrant further investigation.

Reviewer #3 (Remarks to the Author)

In this paper on serine synthesis regulation in cancer, Luo et al. describe how F-box-only protein 7 (FBXO7), an E3 ubiquitin ligase, inhibits serine synthesis by promoting the ubiquitin-proteasome degradation of protein arginine methyltransferase 1 (PRMT1) in hepatocellular carcinoma (HCC). Previously, they had already shown that PRMT1 is upregulated in HCC. PRMT1 methylates phosphoglycerate dehydrogenase (PHGDH) at arginine 236, thereby activating PHGDH and promoting serine synthesis. Nevertheless, the mechanisms underlying overexpression of PRMT1 and regulation of the PRMT1-PHGDH axis in HCC remains unknown.

Here, the authors unravel the details of this mechanism. In a series of elegant experiments including all relevant controls, the authors convincingly show that FBXO7 directly interacts with PRMT1, leading to the lysine 37 ubiquitination and subsequent degradation of PRMT1. FBXO7-mediated PRMT1 downregulation then decreases the extent to which PHGDH is methylated on its target arginine residue, resulting in impaired serine synthesis, accumulation of reactive oxygen species (ROS) levels, and inhibition of HCC cell growth.

The authors have gathered a substantial amount of evidence to support these claims, including in vitro biochemical experiments and clinical data, further strengthening their hypothesis.

Overall, the design of the experiments and the levels of detail are very solid. The data convincingly show that FBXO7 suppresses PHGDH R236 methylation and catalytic activity by downregulating PRMT1 in HCC cells.

I have no major comments, only a few minor comments listed below in bullet points:

- P5: The authors state that “[...] we and others have revealed that PRMT1 is frequently

overexpressed in HCC to promote tumor growth [...]”. Is it really overexpressed or is it just degraded to a lesser extent upon expression? As the mRNA levels seem not to change, I assume it is the latter.

Response: Thanks for your valuable comments. In this study, we found that PRMT1 is upregulated in HCC due to a lesser extent of degradation. Therefore, we have corrected the statement “PRMT1 is overexpressed” as “PRMT1 is upregulated” (Page 2, Line 8; Page 3, Lines 26-27; Page4, Line 3; Page 4, Lines 24-26; Page 8, Line 12; Page 14, Line 18; Page 15, Line 5 in the revised manuscript with track changes).

- P5: Although the authors have used a smart *in silico* screening to identify interaction partners that ultimately provided them the link to FBXO7, I’m wondering why the authors have not performed an unbiased interaction screen based on affinity purification coupled to mass spectrometric identification of interaction partners themselves. The advantage would be that PRMT1 interactions in the relevant cell system (Huh7 and PLC/PRF/5) could be studied this way. I’m not sure whether BioGRID contains interactome data for this specific cell line.

Response: Following your kind suggestion, we performed label-free quantitative proteomics analysis to identify the interaction partners of PRMT1 in Huh7 cells. In line with *in silico* screening results, we found that FBXO7 was the only E3 ligase among the top 10 PRMT1-interacting candidates (Supplementary Fig. 2a; Page 5, Lines 19-22 in the revised manuscript with track changes). We therefore selected FBXO7 for subsequent co-IP validation.

a

Proteins	Ratio (FLAG-PRMT1/Vector)	No. Unique peptides
PRPH	522.78	10
TUSC1	226.47	6
FBXO7	184.09	7
NEFL	176.61	11
TAF15	140.66	15
ANGEL2	138.18	12
RAMAC	133.72	6
HNRNPH3	121.27	10
RBFOX2	110.34	6
LSM14A	109.89	13

Supplementary Figure 2. a FLAG-PRMT1 was immunopurified by FLAG beads in Huh7 cells stably expressing FLAG-PRMT1. Immunoprecipitates were then digested by trypsin and subjected to LC-MS/MS for label-free quantitative proteomics analysis.

- P7: “[...] as shown in Figure 2A, the upregulated protein level of PRMT1 caused by FBXO7 KD was not observed in MG132-treated HCC cells [...]”. The Western blots for the PLC/PRF/5 cells are convincing in this respect, but the Huh7 less so. It’s hard to see the lack of upregulated levels in the MG132 treated FBXO7 KD cells. Have the authors considered more quantitative approaches to determine PRMT1 and FBXO7 levels, such as quantitative targeted mass spectrometry (PRM)?

Response: Thanks for your critical comments. We quantitated the blot intensity of PRMT1 in Huh7 cells using Image J software (below the blots), which showed that PRMT1 level was elevated by knockdown of FBXO7 in Huh7 cells, and this elevation of PRMT1 protein level was not observed in the presence of MG132 treatment (Fig. R3). The less convincing blots might be due to the dumbbell-shaped blots shown in Huh7 cells.

Figure R3. Immunoblotting analysis of PRMT1 and FBXO7 in *FBXO7* KD cells treated with or without MG132 (25 μ M, 6 h).

To this end, we repeated the immunoblotting analysis and found that the upregulated protein level of PRMT1 caused by *FBXO7* KD was indeed not observed in MG132-treated Huh7 cells (Fig. 2a). This observation was further confirmed using another proteasomal inhibitor, bortezomib, in both Huh7 and PLC/PRF/5 cells (Supplementary Fig. 3h). Moreover, although the PRMT1 level was not determined by PRM, we used two different commercial PRMT1 antibodies (Abcam, ab190892; Proteintech, 11279-1-AP) for immunoblotting analysis to exclude the effect of nonspecific recognition of PRMT1 antibodies. Consistently, the upregulation of PRMT1 protein level caused by *FBXO7* KD was not observed in MG132- or bortezomib-treated HCC cells, as evidenced by immunoblotting analysis using two different commercial PRMT1 antibodies (Supplementary Fig. 3g, h) (Page 6, Lines 23-28; and Page 7, Line 1 in the revised manuscript with track changes). These data indicate that FBXO7-mediated downregulation of PRMT1 protein level is attributed to ubiquitin-proteasome degradation.

Figure 2. a Immunoblotting analysis of PRMT1 (anti-PRMT1 antibody: Abcam, ab190892) and FBXO7 in *FBXO7* KD cells treated with or without MG132 (25 μ M, 6 h).

Supplementary Figure 3. g Immunoblotting analysis of PRMT1 (anti-PRMT1 antibody: Proteintech, 11279-1-AP) and FBXO7 in *FBXO7* KD cells treated with or without MG132 (25 μ M, 6 h). **h** Immunoblotting analysis of PRMT1 (anti-PRMT1 antibody: Abcam, ab190892; Proteintech, 11279-1-AP) and FBXO7 in *FBXO7* KD cells treated with bortezomib (500 nM, 5 h).

- P8: In Fig 3C, although the high MW smear as observed in the WT lane is certainly not observed in the K37R cells, the intensity of the bands / smear in the HA IB seems to be increased in the K37R mutant with respect to the WT (minus FLAG-FBXO7). Do the authors have an explanation for this? Also, even though the ubiquitination of the protein results in a smear on the blot, distinct bands are observed. Is it possible, based on these results, to hypothesize on the substrate being mono- or poly-ubiquitinated?

Response: Thank you for your critical comments. The increased intensity of the HA IB band in K37R group (compared with WT minus FLAG-FBXO7 group) is due to the occurrence of a distinct band around 100 kDa, which might be a non-specific band caused by using different batches of GFP antibody (Abcam, ab1218) for co-IP analysis. In this regard, we repeated this experiment using another GFP antibody (Roche,

11814460001) for co-IP analysis. As shown in Fig. 3c in the revised version, reconstituted expression of FLAG-FBXO7 in *FBXO7* KD Huh7 cells obviously elevated the ubiquitination level of wild-type GFP-PRMT1. The increase in GFP-PRMT1 ubiquitination level caused by reconstituted expression of FLAG-FBXO7 could be observed in wild-type, K82R, K202R, and K325R mutants of GFP-PRMT1, but not in K37R mutant, suggesting that K37 is a major ubiquitination site of PRMT1 mediated by FBXO7.

Figure 3. c GFP-PRMT1 (WT or indicated KR mutants) was co-expressed with HA-Ub in *FBXO7* KD Huh7 cells rescued with or without FLAG-FBXO7. After MG132 (25 μ M, 6 h) treatment, IP was performed with GFP antibody, followed by immunoblotting with indicated antibodies.

- In the fragmentation mass spectrum in Fig 3F, no (or, at most, noise intensity level) fragment ion peaks are observed around the Lysine residue with the diGly remnant. However, given the fact that the K-D bond is not cleaved and the total peptide mass corresponds to that of the modified peptides, the data convincingly shows that K37 is ubiquitinated. Was the PHGDH R236 methylation site also confirmed by MS, like the K37 ubiquitination site of PRMT1?

Response: Thank you for your valuable comments. The R236 methylation site for

PHGDH has been confirmed by MS in Fig. 2b in our previous study (Wang *et al.*, *Nat Commun.* 2023;14(1):1011). Please see the Fig. R3 showing PHGDH R236 methylation site below.

Figure. R3 (Fig. 2b from Wang *et al.*, *Nat Commun.* 2023;14(1):1011). MS identification of R236 mono-methylation of FLAG-PHGDH immunopurified by FLAG beads.

- In Fig 4G, the statistical (in)significance indicators for the shPRMT1 shScramble vs shFBXO7 are missing (idem for Figs 5A and B).

Response: Thanks for your kind remind. We have added the *P* value comparing the two groups in Fig. 4g and Fig. 5a and b (Fig. 4g and Supplementary Fig. 4d, e in the revised manuscript).

Figure 4. g Endogenous PHGDH was immunoprecipitated in FBXO7 and/or PRMT1 KD cells, followed by measurement of PHGDH activity.

Supplementary Figure 5. d, e Total serine (**d**) and glycine (**e**) levels in *FBXO7* and/or *PRMT1* KD cells cultured in serine- and glycine-depleted (-SG) medium.

- Have the authors considered including experiments with a mutant PHGDH that lacks the R236 target methylation site? If so, why were these not included in the manuscript?

Response: Thank you for your critical comments. Two mutants of PHGDH (R236K mutant that fails to be methylated by PRMT1, and V83A mutant that cannot interact with PRMT1) were used in our previous study (Wang *et al.*, *Nat Commun.* 2023;14(1):1011). In our previous study, by using PHGDH R236K and V83A mutants, we have well demonstrated that PRMT1-mediated R236 methylation elevates PHGDH activity, promotes serine synthesis, ameliorates oxidative stress, and supports HCC growth (Figure 2, 3, 4, and 5 in *Nat Commun.* 2023;14(1):1011). Therefore, in this current study, we focused on investigating the mechanisms underlying upregulation of PRMT1 and regulation of PRMT1-PHGDH axis in HCC, without using PHGDH mutants. Instead, PRMT1 K37R mutant was used in the current study to investigate its ubiquitination mechanism and biological effects (Fig. 3, 4, 5, 6 in the current study).

- The model in Fig 7M suggests that low levels of FBXO7 lead to ROS accumulation (I suppose that the spiked balloon with the text 'ROS' means 'accumulation of ROS'). However, in Fig 5H, the authors show that the depletion of FBXO7 causes a reduction of ROS levels and that the depletion of PRMT1 leads to increased ROS levels. Also,

ectopic FBXO7 expression promotes ROS accumulation in control cells and has no obvious effect in PRMT1 KD cells (Fig 5I). This suggests that FBXO7 induces oxidative stress by downregulating PRMT1 in HCC cells. Therefore, the description of the results and the cartoon in Fig 7M seem to contradict each other. Could the authors comment on that?

Response: In the previous version of Fig. 7m, ROS accumulation was inhibited (see the blunt arrow) in “FBXO7 low” conditions. We apologize for this misunderstanding incurred. We therefore simplified this schematic model to make it clearer that FBXO7 promotes ubiquitin-mediated degradation of PRMT1, leading to decreased PHGDH activity and increased ROS levels in HCC cells.

Previous Figure 7m.

Revised Figure 7m (Supplementary Fig. 8 in the revised version). Schematic model for the mechanism of FBXO7-PRMT1-PHGDH axis in regulating serine synthesis, oxidative stress, and HCC growth. The E3 ubiquitin ligase FBXO7 promotes PRMT1 degradation by

lysine 37 ubiquitination, leading to reduction of PHGDH arginine methylation and catalytic activity, suppression of serine synthesis, accumulation of ROS, and inhibition of HCC growth.

Reviewer #4 (Remarks to the Author):

This manuscript entitled "FBXO7 Ubiquitinates PRMT1 to Suppress Serine Synthesis and Tumor Growth in Hepatocellular Carcinoma" by Li Luo et al identified a tumor-suppressive role of FBXO7 in HCC. The authors first showed that FBXO7 is a PRMT1-interacting partner by in vitro cell line-based assay. Next they demonstrated that FBXO7 ubiquitinate PRMT1 and identified the K37 in PRMT1 is the key ubiquitination site. Further functional studies showed that FBXO7 suppressed PHGDH activity by downregulating PRMT1-mediated PHGDH methylation (R236me1), causing the inhibition of serine synthesis and HCC repression. Overall, the manuscript is well-designed and organized, the evidence supports the conclusion, and the findings will be of broad interest to the Liver cancer field.

Major concern:

1. in Figure 1I, by IP assay, it seems the PRMT1 protein catalytic fragment (23-162) is the key for the interaction with FBXO7, however, this is not precisely demonstrated in the manuscript. In the Fig.1K, which part of FBXO7 is required for its interaction with PRMT1? this is not well defined in its current form. Moreover, it would be good to show the co-localization for their interaction, e.g. by immunofluorescent staining as a surrogate.

Response: Thank you for your critical comments and valuable suggestions. We added two truncation mutants (23-162 and Δ 23-162) of GFP-PRMT1 for IP analysis, and found that the 23-162aa truncate (catalytic domain) of GFP-PRMT1 retained its binding with FLAG-FBXO7, whereas deletion of the catalytic domain (Δ 23-162) failed to interact with FLAG-FBXO7. These data further confirmed that the catalytic domain (23-162aa) of PRMT1 was required for its binding with FBXO7 (Fig. 1h, j).

We also added two truncation mutants ($\Delta 181-324$ and $325-522aa$) of FLAG-FBXO7 to investigate the binding domain of FBXO7 required for PRMT1 interaction. As shown in Fig. 1k, m, either deletion of UBL domain (named 79-522aa) or FB domain (named $\Delta 181-324$) of FLAG-FBXO7 markedly decreased its interaction with GFP-PRMT1, whereas deletion both UBL and FB domains of FLAG-FBXO7 (named $325-522aa$) failed to interact with GFP-PRMT1. These results indicate that the UBL (1-78aa) and FP (180-324aa) domains of FBXO7 were required for its binding with PRMT1.

Figure 1. **h** Schematic representation of full-length (FL) PRMT1 and different truncation mutants. **i, j** GFP-PRMT1 FL or indicated truncation mutants were co-expressed with FLAG-FBXO7 in HEK293T cells. FLAG-FBXO7 was immunoprecipitated with FLAG beads, followed by immunoblotting analysis of GFP-PRMT1 using GFP antibody. **k** Schematic representation of full-length (FL) FBXO7 and different truncation mutants. UBL, ubiquitin-like domain; FP, FBXO7-PI31 dimerization domain; PRR: proline-rich region. **l, m** FLAG-FBXO7 FL or indicated truncation mutants were co-expressed with GFP-PRMT1 in HEK293T cells. FLAG-FBXO7 was immunoprecipitated with FLAG beads, followed by immunoblotting analysis of GFP-PRMT1 using GFP antibody.

Following your suggestion, we also performed immunofluorescent staining to examine the colocalization of FBXO7 with PRMT1. As shown in Supplementary Fig. 2f, FBXO7 was colocalized with PRMT1 in both Huh7 and PLC/PRF/7 cells, further confirming the interaction of FBXO7 with PRMT1 (Page 6, Lines 1-3 in the revised manuscript with track changes).

Supplementary Figure 2. f Immunofluorescent analysis of the colocalization of FLAG-FBXO7 with GFP-PRMT1 in Huh7 and PLC/PRF/5 cells. Scale bars: 10 μ m.

2. Although the authors provided both the negative correlation and xenograft studies using clinical biopsies samples, further experimental demonstration of reciprocal interaction between FBXO7 and PRMT1, and perhaps the presence of PRMT1 ubiquitination site, in primary HCC tumors will strengthen the conclusion.

Response: Thanks for your valuable suggestions. We analyzed the binding of FBXO7 with PRMT1 in 6 HCC tissues using reciprocal co-IP analysis, and found that FBXO7 interacted with PRMT1 in human HCC tissues (Fig. 7k). Although the presence of PRMT1 ubiquitination site was not measured in HCC tissues due to the lack of site-specific antibody recognizing PRMT1 K37 ubiquitination, we measured the ubiquitination level of PRMT1 in these 6 HCC tissues using co-IP analysis. As shown in Fig. 7k, PRMT1 ubiquitination was observed in all detected HCC tissue samples. Interestingly, the PRMT1 ubiquitination level was positively correlated with the level of FBXO7-PRMT1 interaction (Fig. 7l) (Page 12, Line 19-21 in the revised manuscript with track changes). These data further indicate that FBXO7 interacts with PRMT1 and

ubiquitinates it in HCC tissues.

Figure 7. k IP and immunoblotting analysis with indicated antibodies in 6 HCC tissues (T1-T6). **l** Pearson correlation test analyzing the relationship between the levels of FBXO7-PRMT1 interaction and PRMT ubiquitination in HCC tissues based on band intensity in **k** ($n = 6$ samples).

Minor concerns:

P6, "Figure 2H and 2I)" should allude to Fig.1H,1I.

P7, Figure 2J and 2K should mention "Figure 1J and 1K"

Response: We apologize for these mistakes, and have corrected them in the revised manuscript.

P34, in Figure 3C: top panel: I can see GFPtagged PRMT1 was ubiquitinated as visualized by the smear bands in second WT and K82R/202R/325R samples. However, in the bottom IB:GFP loading control, the GFP bands showed as distinctive single bands (70 kD), should not they appear as smear bands?

Response: Thanks for your critical comments. We repeated this experiment in Fig. 3c to show GFP WCL bands with longer exposure time. As shown below, the GFP WCL bands with long exposure time indeed appeared as smear bands, instead of distinctive single bands. However, because the enforced expression of extrinsic GFP was quite strong and the signal of the smear bands was extreme weak, they cannot truly reflect the increased or reduced ubiquitination levels in different groups. Therefore, GFP-PRMT1 protein in cell lysates was pulled down and enriched by GFP antibody, then the ubiquitinated GFP-PRMT1 level could be easily measured by HA antibody (HA-Ub) in the enriched immunoprecipitates (Fig. 3c).

Figure 3. c GFP-PRMT1 (WT or indicated KR mutants) was co-expressed with HA-Ub in *FBXO7* KD in Huh7 cells rescued with or without FLAG-FBXO7. After MG132 (25 μ M, 6 h) treatment, IP was performed with GFP antibody, followed by immunoblotting with indicated antibodies.

P34, in Figure 3E: Please elaborate how the band intensity was quantified, because it seems to me the data differences at 6/12/24 hr were not that different as presented.

Response: Thanks for your critical comments. The band intensity was quantified using Image J software. The band intensity at each time point (6 h, 12 h, 24 h) was then normalized to that at 0 h for each group. The original data were shown as follows:

WT		shScramble				shFBX07 #2			
CHX (h)	0	6	12	24	0	6	12	24	
	13154.669	8053.719	4207.719	1530.284	16240.134	13396.134	12137.255	8319.497	
	11080.134	5170.891	2553.648	1624.93	15387.719	12178.719	10809.426	6846.376	
	10839.79	6915.134	4024.255	1899.284	16723.841	12568.548	13137.962	7966.376	
Normalize to 0 h (%)	100	61.2232737	31.9865061	11.6330103	100	82.4878292	74.7361752	51.2280071	
	100	46.6681269	23.0470859	14.6652559	100	79.145707	70.2470977	44.4924683	
	100	63.7939849	37.1248428	17.5214095	100	75.153477	78.558281	47.6348466	
K37R		shScramble				shFBX07 #2			
CHX (h)	0	6	12	24	0	6	12	24	
	23782.569	17963.912	16942.497	10985.205	20678.326	16707.619	15209.033	9349.619	
	26205.104	18665.276	17722.276	11210.518	24750.347	18502.711	18034.539	12801.175	
	24336.861	19616.912	19867.79	10010.498	21279.74	16373.619	17171.619	10118.619	
Normalize to 0 h (%)	100	75.5339425	71.2391374	46.190153	100	80.7977348	73.5506008	45.2145836	
	100	71.2276357	67.6291	42.7799027	100	74.7573802	72.8658026	51.7211941	
	100	80.6057609	81.636617	41.1330697	100	76.9446384	80.6946842	47.5504823	

REVIEWERS' COMMENTS

Reviewer #1 (Remarks to the Author):

Thank you for asking me to review the revised manuscript 'FBXO7 Ubiquitinates PRMT1 to Suppress Serine Synthesis and Tumor Growth in Hepatocellular Carcinoma' by Luo et al. The authors have responded to the various reviewers' comments and have added a significant amount of new data. These data have significantly strengthened the manuscript and the confidence in their conclusions.

The work adding the functional analysis of the K37R mutant in vitro and the effect on tumour growth in vivo is to be commended.

I believe that the manuscript can now be published.

Reviewer #2 (Remarks to the Author):

The revised version of the manuscript reflects a commendable effort by the authors. They have addressed the comments and especially resolved the previous concerns regarding the biological significance. The observed difference in tumor growth among mice fed with an S/G- diet is remarkable and compelling.

One minor aspect that warrants attention is the potential impact of FBX protein stable overexpression on the cell viability. Should such overexpression induce ROS accumulation and subsequent cell death, the surviving cells must have undergone metabolic adaptations. Consequently, this raises concern on the relevance of this experimental model for addressing the questions. It is recommendable to possibly address this or at least elaborate on this point in the discussion.

Reviewer #3 (Remarks to the Author):

I think that the authors have adequately addressed all of my questions and comments in their rebuttal letter and have adapted the manuscript accordingly. Therefore, I have no further comments.

Reviewer #4 (Remarks to the Author):

My questions were all addressed!

Point-by-point response

REVIEWERS' COMMENTS

Reviewer #1 (Remarks to the Author):

Thank you for asking me to review the revised manuscript 'FBXO7 Ubiquitinates PRMT1 to Suppress Serine Synthesis and Tumor Growth in Hepatocellular Carcinoma' by Luo et al. The authors have responded to the various reviewers' comments and have added a significant amount of new data. These data have significantly strengthened the manuscript and the confidence in their conclusions.

The work adding the functional analysis of the K37R mutant in vitro and the effect on tumour growth in vivo is to be commended.

I believe that the manuscript can now be published.

Response: Thanks for your valuable comments.

Reviewer #2 (Remarks to the Author):

The revised version of the manuscript reflects a commendable effort by the authors. They have addressed the comments and especially resolved the previous concerns regarding the biological significance. The observed difference in tumor growth among mice fed with an S/G- diet is remarkable and compelling.

One minor aspect that warrants attention is the potential impact of FBX protein stable overexpression on the cell viability. Should such overexpression induce ROS accumulation and subsequent cell death, the surviving cells must have undergone metabolic adaptations. Consequently, this raises concern on the relevance of this experimental model for addressing the questions. It is recommendable to possibly address this or at least elaborate on this point in the discussion.

Response: Thanks for your valuable suggestion. When HCC cells were cultured in serine- and glycine-replete medium, FBXO7 overexpression only slightly inhibited cell

growth mainly due to the metabolic adaptation caused by increased import of exogenous serine and glycine. However, when serine and glycine were deprived, FBXO7 overexpression markedly induced ROS accumulation and cell death. In this case, a very small portion of HCC cells still survived, which might be attributed to the activation of salvage pathways for GSH and NADPH synthesis or GSH- and NADPH-independent antioxidant pathways. We have elaborated this point in Discussion section (Page 13, Lines 26-28; Page 14, Line 1).

Reviewer #3 (Remarks to the Author):

I think that the authors have adequately addressed all of my questions and comments in their rebuttal letter and have adapted the manuscript accordingly. Therefore, I have no further comments.

Response: Thanks for your valuable comments.

Reviewer #4 (Remarks to the Author):

My questions were all addressed!

Response: Thanks for your valuable comments.